# VASP-mediated actin dynamics activate and recruit a filopodia myosin

**Ashley L Arthur[1†], Amy Crawford[1], Anne Houdusse[2], Margaret A Titus[1*]**

[1]Department of Genetics, Cell Biology, and Development, University of Minnesota, Minneapolis, United States; [2]Structural Motility, Institut Curie, Paris Université Sciences et Lettres, Sorbonne Université, Paris, France

**Abstract** Filopodia are thin, actin-based structures that cells use to interact with their environments. Filopodia initiation requires a suite of conserved proteins but the mechanism remains poorly understood. The actin polymerase VASP and a MyTH-FERM (MF) myosin, DdMyo7 in amoeba, are essential for filopodia initiation. DdMyo7 is localized to dynamic regions of the actin-rich cortex. Analysis of VASP mutants and treatment of cells with anti-actin drugs shows that myosin recruitment and activation in *Dictyostelium* requires localized VASP-dependent actin polymerization. Targeting of DdMyo7 to the cortex alone is not sufficient for filopodia initiation; VASP activity is also required. The actin regulator locally produces a cortical actin network that activates myosin and together they shape the actin network to promote extension of parallel bundles of actin during filopodia formation. This work reveals how filopodia initiation requires close collaboration between an actin-binding protein, the state of the actin cytoskeleton and MF myosin activity.

**\*For correspondence:**
titus004@umn.edu

**Present address:** [†]Department of Molecular Biophysics & Biochemistry, Yale University, New Haven, United States

**Competing interests:** The authors declare that no competing interests exist.

## Introduction

The efficient and directed migration of cells depends on their ability to detect and respond to chemical signals and physical cues in the environment. Filopodia are dynamic, thin membrane projections supported by a parallel bundle of actin filaments. They detect extracellular cues and play roles in processes such as neuronal growth cone guidance, durotaxis, cell-cell junction formation during development, and metastasis (*Heckman and Plummer, 2013*; *Arjonen et al., 2011*; *Cao et al., 2014*; *Shibue et al., 2012*; *Gallop, 2020*). Although most intensely studied in animal cells, filopodia are ubiquitous in moving cells and have been observed in various Rhizaria, including predatory vampire amoebae, Discoba, Apusoza, Amoeboza, and Holozoa (*Sebé-Pedrós et al., 2013*; *Cavalier-Smith and Chao, 2003*; *Hess et al., 2012*; *Hanousková et al., 2019*; *Yabuki et al., 2013*). Filopodia formation is orchestrated by a conserved core set of proteins that drive the formation and extension of actin bundles. These include a Rho family GTPase (Rac1, Cdc42), an actin polymerase (VASP or formin), an actin cross-linker and a MyTH4-FERM Myosin (MF; **my**osin **t**ail **h**omology 4, band **4**.1, **e**zrin, **r**adixin, **m**oesin) (*Mattila and Lappalainen, 2008*; *Sebé-Pedrós et al., 2013*; *Nobes and Hall, 1995*; *Tuxworth et al., 2001*; *Faix et al., 2009*). MF myosin motors regulate the formation of filopodia and other parallel actin based structures (*Weck et al., 2017*). *Dictyostelium* amoebae null for DdMyo7 do not produce filopodia or any filopodia-like protrusions, and expression of Myo10 in various mammalian cell types induces filopodia formation, implicating these myosins in filopodia initiation (*Bohil et al., 2006*; *Tuxworth et al., 2001*; *Sousa and Cheney, 2005*). These MF myosins are strikingly localized to filopodia tips, yet their function during filopodia initiation remains poorly defined.

Filopodia are slender, actin-filled projections that extend from the cell cortex. Actin polymerization aided by VASP or formin facilitates the formation of a critical bundle size of 15–20 filaments which overcomes membrane tension and extends outwards as the actin filaments are cross-linked

together (*Mattila and Lappalainen, 2008*; *Mogilner and Rubinstein, 2005*). Two models have been proposed for how these parallel actin bundles are formed. In one case, the branched actin network is reorganized into a parallel array by the polymerase and bundler VASP (*Svitkina et al., 2003*; *Yang and Svitkina, 2011*). In an alternative model, a linear actin polymerase such as formin nucleates new filaments that are rapidly bundled together and grow perpendicular to the plasma membrane (*Faix and Rottner, 2006*). MF myosins are thought to act during initiation by cross-linking actin filaments (*Tokuo et al., 2007*), perhaps zipping them together as the motors walk up the filament toward the cortex (*Berg and Cheney, 2002*; *Ropars et al., 2016*). Support for this model comes from the observation that forced dimers of motors can induce filopodia or filopodia-like protrusions in cells (*Tokuo et al., 2007*; *Arthur et al., 2019*; *Masters and Buss, 2017*; *Liu et al., 2021*). Myo10 has also been implicated in elongation by transporting VASP toward the tip of the growing filopodium to promote continued growth (*Tokuo and Ikebe, 2004*).

The mechanism by which MF myosins are recruited to filopodial initiation sites is not well-understood. These myosins are regulated by head-tail autoinhibition, with the myosin folded into a compact conformation whereby binding of the C-terminal MF domain to the motor domain inhibits its activity. Opening up of the myosin followed by dimerization is required for activation of the myosin (*Umeki et al., 2011*; *Sakai et al., 2011*; *Yang et al., 2006*; *Arthur et al., 2019*). Partner binding mediated by their MyTH4-FERM (MF) domains can typically stabilize the open, activated form of these myosins as well as promote dimerization (*Sakai et al., 2011*; *Arthur et al., 2019*; *Liu et al., 2021*). The MF domains in these myosins can indeed mediate interaction with partner proteins such as microtubules (*Weber et al., 2004*; *Toyoshima and Nishida, 2007*; *Planelles-Herrero et al., 2016*) and the cytoplasmic tails of adhesion and signaling receptors (*Hirao et al., 1996*; *Hamada et al., 2000*; *Zhang et al., 2004*; *Zhu et al., 2007*; *Pi et al., 2007*). Myo10 has a unique tail among MF myosins with PH domains that bind to PIP3 rich membranes, which facilitates autoinhibition release and subsequent dimerization via a coiled-coil domain (*Plantard et al., 2010*; *Umeki et al., 2011*; *Lu et al., 2012*; *Ropars et al., 2016*). DdMyo7, like mammalian microvilli and stereocilia myosins Myo7 and Myo15, lacks PH domains and it is not clear if activation is regulated by the concentration of some partners, or additional cellular signals.

MF myosin and the actin regulators VASP and formin have been observed to coalesce into punctae during the initiation step that precedes extension (*Petersen et al., 2016*; *Young et al., 2018*; *Cheng and Mullins, 2020*). The mechanism by which Myo7s are targeted to such sites is unknown, but the first step of this process is likely to be regulated via relief of autoinhibition. Characterization of filopodia formation in *Dictyostelium* allows for a systematic examination of how DdMyo7 is recruited to the cell cortex through identification of the essential features of this myosin. The mechanism of DdMyo7 recruitment to the cortex and its potential functional relationship with the actin polymerase and bundler VASP were investigated to gain new insights into motor activation and the early steps of filopodia formation.

## Results

### The DdMyo7 motor restricts cortical localization of the tail

Disruption of the *Dictyostelium myoI* (*myo7*) gene encoding DdMyo7 results in a significant defect in filopodia formation that is rescued by expression of GFP-DdMyo7 (*Figure 1A*, *Figure 1—figure supplements 1A* and *2A*; *Tuxworth et al., 2001*; *Petersen et al., 2016*). Most strikingly localized to filopodia tips, DdMyo7 is also in the cytosol and localized to the leading edge of cells (*Figure 1B*, *Figure 1—figure supplement 1A*). In the course of characterizing functionally important regions of DdMyo7, it was observed that the tail domain (aa 809 - end) is localized all around the cell periphery in contrast to the full-length myosin which is often restricted at one edge of the cell cortex, (*Figure 1B*; *Petersen et al., 2016*; *Arthur et al., 2019*). This was unexpected as the myosin tail region is largely regarded as playing a key, even determining, role in targeting myosins. DdMyo7-mCherry and GFP-DdMyo7 tail were co-expressed in *Dictyostelium* cells (*Figure 1B*, *Figure 1—figure supplement 1B*) and a line scan through the cell showed that while both are present at a region of the cell that is extending outwards and producing filopodia (i.e. the leading edge), the tail is also strongly enriched in the cell rear (*Figure 1C*, line from *Figure 1B*). The extent of co-localization of the full-length myosin with the tail domain was assessed using cytofluorograms. This method

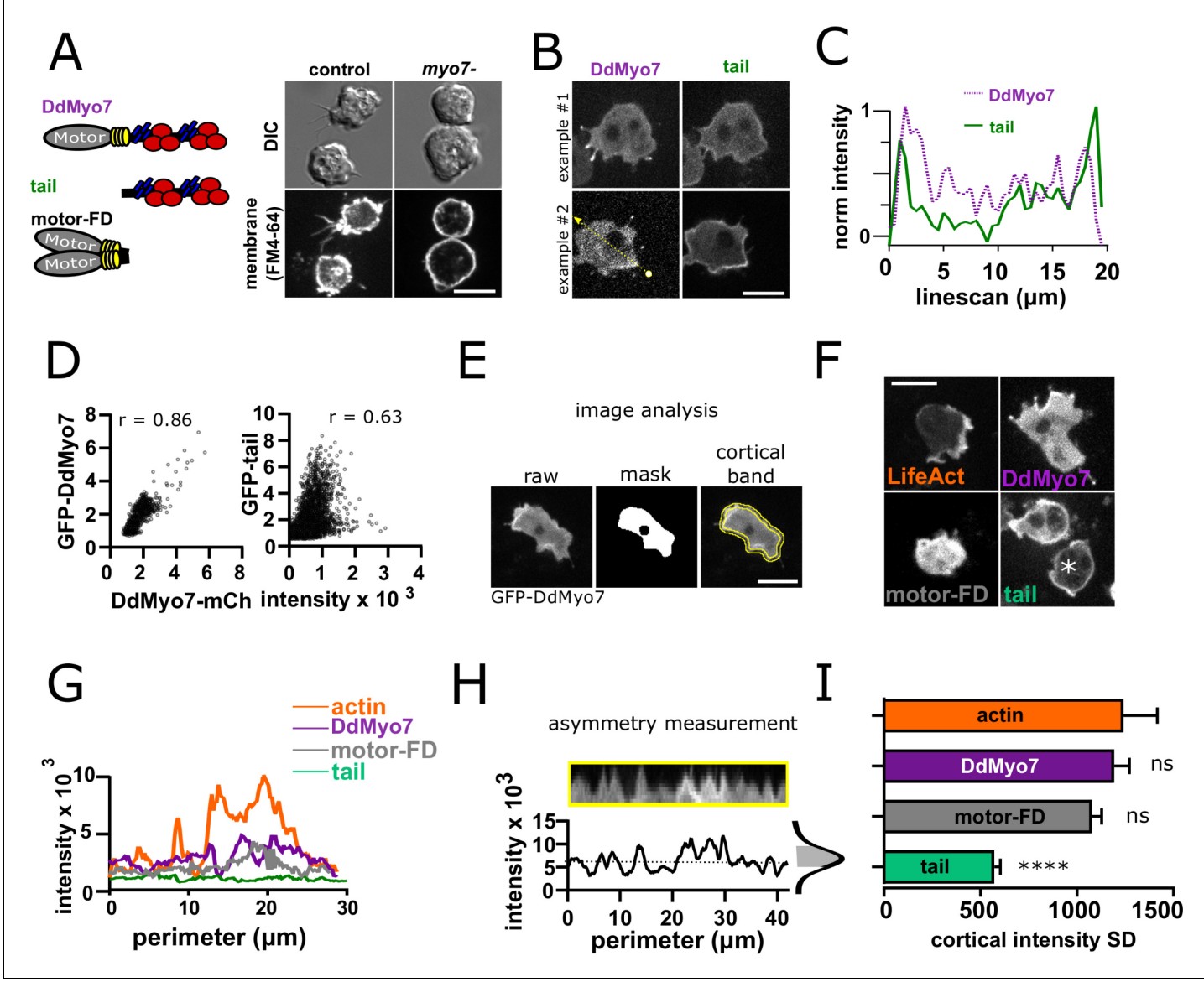

**Figure 1.** DdMyo7 has a distinct cortical localization from its tail domain. (**A**) (left) Schematic of DdMyo7 illustrating its motor domain (gray), 4 IQ domains (yellow) and tandem MyTH4-FERM domains (blue-MyTH4, red-FERM) in the tail, the tail fragment, and a motor forced dimer (motor-FD); (right) *Dictyostelium* control, or *myo7* null cells visualized with DIC and the membrane dye FM4-64 showing DdMyo7 is critical for filopodia formation. (**B**) Confocal images showing two examples of wild-type cells co-expressing DdMyo7-mCherry and GFP-DdMyo7-tail. The localization of DdMyo7-mCherry is at the cortex and in filopodia tips, and GFP-tail fragment localized around cortex. (**C**) Line intensity profile along the line shown in panel B. (**D**) Cytofluorograms of a representative field of cells comparing the colocalization between DdMyo7-mCherry intensity (x-axis) and GFP-DdMyo7 or GFP-DdMyo7 tail intensity (y-axis). (**E**) Analysis strategy for measuring entire cell peripheral intensity. (**F**) Micrographs of cells expressing RFP-Lifeact, GFP-DdMyo7, GFP-Tail, or GFP-Motor-Forced Dimer (FD) asterisks (*) on GFP-tail indicates the cell analyzed in G. (A,B,E,F) Scale bars are 10 µm. (**G**) Peripheral line scan intensity of cells from F. (**H**). Sample cortical band intensity showing the mean and variation of intensities around the periphery (asymmetry measurement), shaded region of the intensity distribution represents the standard deviation. (**I**). Cortical band standard deviation (SD; n > 93 cells from three experiments for each group) (see also *Figure 1—source data 1*). A higher SD indicates asymmetric localization. One-way ANOVA with multiple comparison correction compared to actin, ****p<0.001, ns not significant (see also *Figure 1—source data 2*). The online version of this article includes the following source data and figure supplement(s) for figure 1:

**Source data 1.** Values of the cortical standard deviation measurements (cortical asymmetry) for each cell for *Figure 1I*.
**Source data 2.** Statistical test results for *Figure 1I*.
**Figure supplement 1.** DdMyo7 is localized to filopodia and required for their formation.
**Figure supplement 2.** Analysis of protein expression.

quantifies co-localization of two proteins by comparing the intensity of the two fluorescent signals on a pixel-wise basis for a field of cells with *r* representing the correlation coefficient (*Bolte and Cordelières, 2006*). Quantification revealed less correlation between DdMyo7 and the tail (r = 0.63) than is seen for the two full-length myosins (r = 0.86) (*Figure 1D*). Due to the transient polarity of vegetative cells, a second measure of distribution of DdMyo7 on the cortex was carried out. To quantify the localized enrichment of DdMyo7 without any bias for manually identifying the leading edge, the intensity all around the cortex of cells was measured using a high throughput automated image analysis FIJI macro, *Seven* (*Petersen et al., 2016*; *Figure 1E*). Actin is typically enriched at the leading edge with high intensity at one edge and lower intensity elsewhere (*Figure 1F,G*). The standard deviation of intensities values around the periphery was calculated for each cell (as exemplified in *Figure 1H*). This analysis shows that actin (RFP-Lifeact; *Brzeska et al., 2014*) and GFP-DdMyo7 are localized asymmetrically with a high standard deviation of cortical intensity (*Figure 1F, G,I*). In contrast, the tail appears uniformly localized around the periphery (*Figure 1F,G*) and has a lower average SD for the cortical intensity around the cell periphery (*Figure 1I*). Interestingly, a tailless forced dimer of the motor (motor-FD; *Figure 1A*) that is capable of inducing filopodial protrusions (*Arthur et al., 2019*) is also localized asymmetrically with a high standard deviation of intensity (*Figure 1F,I*). Together, these data establish that restricted localization of full-length DdMyo7 is dependent on its motor domain.

## DdMyo7 is localized to dynamic cortical actin

The apparent asymmetrical distribution of DdMyo7 to the leading edge (*Figure 1F*) suggested that the actin at the cortex has a role in the recruitment and localization of DdMyo7. DdMyo7 and actin appear to colocalize at the leading edge in cells expressing both GFP-DdMyo7 and an actin marker (Lifeact) (*Figure 2A*). Linescans show that indeed there is strong correlation of Lifeact and DdMyo7 fluorescence around the cell periphery (*Figure 2B*, yellow line from 2A). Cytofluorogram analysis also revealed strong correlation of the DdMyo7 and actin fluorescence intensities (*Figure 2C*). A line scan taken perpendicular to an extending leading edge showed a steady accumulation of DdMyo7 intensity at the same time as an increase in actin intensity (*Figure 2D,E*). DdMyo7 and actin normalized intensities during leading edge extension were measured for 10 independent cells and a spline fit shows a robust correlation with actin intensity (*Figure 2F*).

The dependence of DdMyo7 localization on actin polymerization at the cortex was tested by treating cells with actin modulating drugs. CytochalasinA (cytoA) binds to the fast-growing (barbed) end of actin filaments and blocks incorporation of actin monomers, capping and stabilizing filaments (*Cooper, 1987*). LatrunculinA (latA) sequesters monomers and prevents actin filament growth, and CK-666 blocks the Arp2/3 complex and actin branching (*Coué et al., 1987*; *Nolen et al., 2009*). Cells were incubated with each of these drugs and the impact on DdMyo7 cortical targeting was assessed by measuring the intensity ratio of DdMyo7 in a 0.8 µm band around the perimeter versus the cytoplasm (see *Figure 1E*). Control cells have a cortex:cytoplasm ratio of about ~1.2, indicating an overall 20% enrichment of DdMyo7 on the cell cortex (*Figure 2G,H*; *Arthur et al., 2019*). Treatment of cells with either cytoA or latA reduced the ratio to ~1 indicating a total loss of cortical localization of DdMyo7 (*Figure 2H*). CK-666 also significantly reduced DdMyo7 cortical recruitment (*Figure 2G,H*). Consistent with the reduction or loss of polymerized actin and cortical DdMyo7, filopodia formation was significantly reduced with these actin modulating drugs (*Table 1*). Cells were also treated with jasplakinolide (jasp) that promotes monomer nucleation and stabilizes ADP-Pi actin filaments (*Merino et al., 2018*). Jasp treatment had the opposite effect, resulting in increased recruitment of DdMyo7 to the cortex and increased filopodia formation (*Figure 2G,H*; *Table 1*). The addition of PI3Kinase inhibitors did not disrupt DdMyo7 cortical targeting or filopodia formation, but did disrupt the localization of a PIP3 reporter (GFP-CRAC, *Parent et al., 1998*; *Figure 2—figure supplement 1A–C*, *Table 1*). Microtubules have been implicated in regulating filopodia in melanoma cells, modulating their localization, density and merging (*Schober et al., 2007*). Treatment of *Dictyostelium* cells with the microtubule depolymerizing compound nocodazole disrupts cytosolic microtubules (visualized by cells expressing GFP-tubulin *Neujahr et al., 1998*, *Figure 2—figure supplement 1E*). Nocodazole reduced DdMyo7 cortical targeting and partially inhibited filopodia formation, particularly at higher concentrations (50 µM) (*Figure 2—figure supplement 1E,G,I*; *Table 1*). Interestingly, cortical targeting of the DdMyo7 tail fragment is not reduced in cells treated with nocodazole (*Figure 2—figure supplement 1F,H*) in spite of the presence of two MF domains

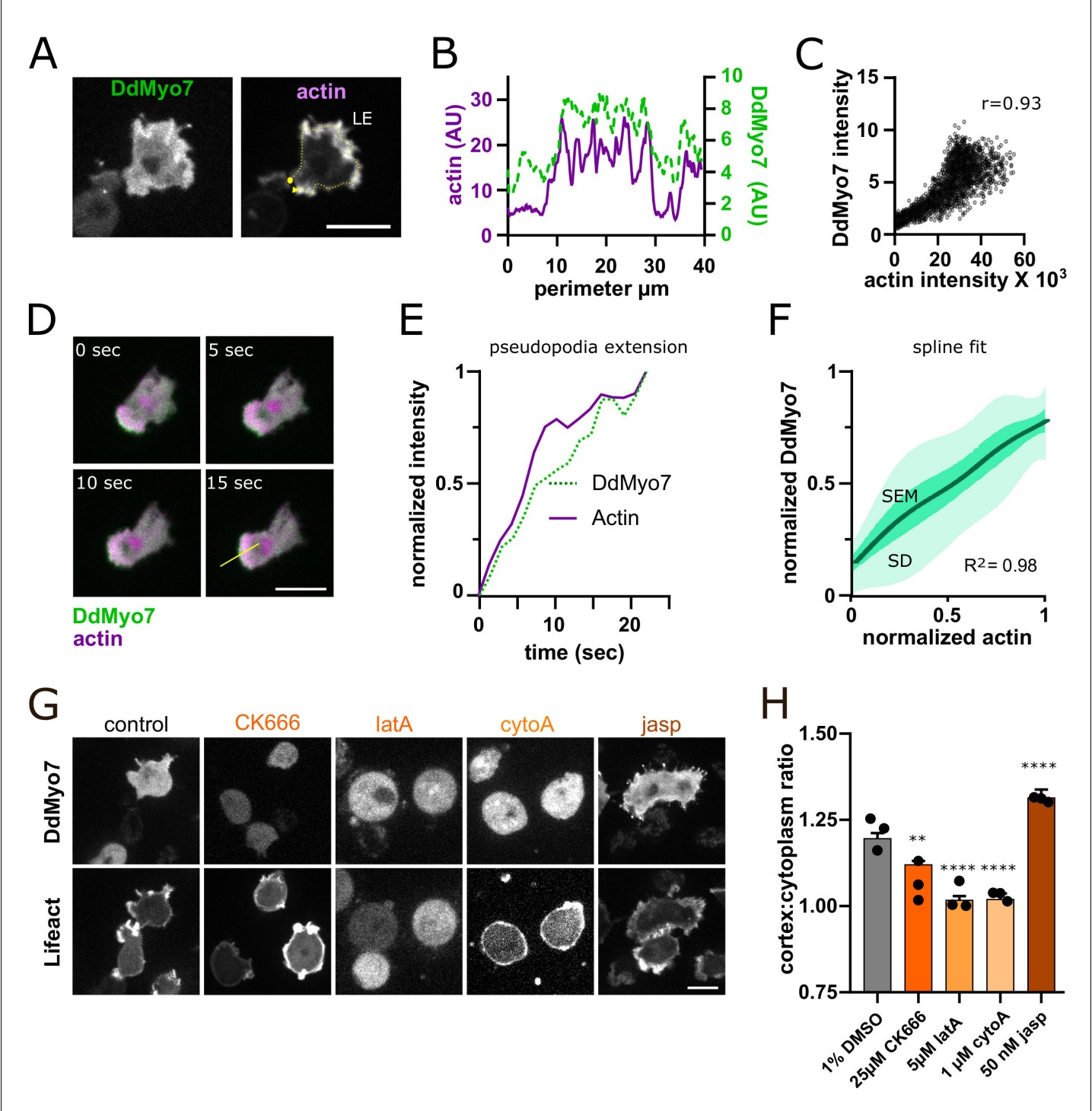

**Figure 2.** Actin dynamics regulate DdMyo7 recruitment to the cortex. (**A**) *Dictyostelium* co-expressing GFP-DdMyo7 and RFP-Lifeact. (**B**) Line intensity profile from yellow dotted line in A (circle = beginning, arrowhead indicates end of scan). (**C**) Cytofluorogram from a representative field of cells showing the colocalization of actin and DdMyo7, r is correlation coefficient. (**D**) Confocal image series of an extending pseudopod. (**E**) Normalized linescan intensity profile of DdMyo7 and actin in extending pseudopod along the line from panel D. (**F**) Intensity correlation of GFP-DdMyo7 and RFP-Lifeact plotted as the average spline fit of 10 extending pseudopodia (SD - light green shading, SEM - dark green shading, see also *Figure 2—source data 1*). (**G**) Confocal micrographs of cells expressing GFP-DdMyo7 (top) or RFP-Lifeact (actin, bottom) treated with specified drug. (**A,D,G**) Scale bar is 10 μm. (**H**) Cortex:cytoplasm ratio (cortex is 0.8 μm band of cell periphery, cytoplasm is the rest of cell excluding the nucleus) of GFP-DdMyo7 of cells treated with anti-actin drugs, circles are experimental means (see also *Table 1*). One-way ANOVA with multiple comparison correction, shown to 1% DMSO control, **$p<0.01$, p****$<0.0001$.

*Figure 2 continued on next page*

*Figure 2 continued*

The online version of this article includes the following source data and figure supplement(s) for figure 2:

**Source data 1.** Normalized intensity values for each of the 10 extending pseudopodia in *Figure 2F*.

**Figure supplement 1.** Effects of microtubule and membrane phospholipid inhibitors on DdMyo7 targeting and filopodia formation.

with micromolar affinity for microtubules in the DdMyo7 tail (*Planelles-Herrero et al., 2016*). Additionally, DdMyo7 is not strongly localized to microtubules (*Figure 2—figure supplement 1D,E*). Together, the data suggest that microtubule-actin crosstalk may impact the dynamics of the cortical actin network and thus indirectly control the cortical recruitment of full-length DdMyo7.

## The role of VASP in DdMyo7 cortical recruitment

Filopodia initiation and extension is driven by regulators of actin polymerization such as VASP and formin (*Mattila and Lappalainen, 2008*; *Figure 3A,B*). The actin bundler/polymerase DdVASP accumulates at the leading edge of cells and is important for filopodia formation (*Han et al., 2002*; *Breitsprecher et al., 2008*). Interestingly, the *Dictyostelium vasp* null mutant phenocopies the *myo7* null mutant - it lacks filopodia, has reduced adhesion and smaller cell size (*Han et al., 2002*; *Tuxworth et al., 2001*). This is of particular note as mammalian Myo10 and VASP are observed to co-transport in filopodia and co-immunoprecipitate, suggesting a role for Myo10 in the transport of VASP to filopodia tips to promote filopodia growth (*Tokuo and Ikebe, 2004*; *Lin et al., 2013*). There is currently no evidence that DdMyo7 and *Dictyostelium* VASP (DdVASP) interact with each other and co-immunoprecipitation experiments in *Dictyostelium* did not detect any interaction (*Figure 3—figure supplement 1A*). Surprisingly, in spite of this lack of interaction, DdMyo7 fails to target efficiently to the cortex of *vasp-* cells (*Figure 3C,D*). In contrast, DdVASP localizes to the cortex regardless of the presence of DdMyo7 (*Figure 3C,D*).

Formins are localized to the leading edge of migrating cells and play an important role in filopodia formation (*Schirenbeck et al., 2005*; *Pellegrin and Mellor, 2005*; *Yang et al., 2007*). Formins promote actin polymerization by incorporating actin monomers at the barbed end of actin filaments and elongate parallel actin filaments (*Breitsprecher and Goode, 2013*; *Mellor, 2010*). The *Dictyostelium* diaphanous related formin dDia2 makes significant contributions to filopodia formation and is required for normal filopodia length (*Figure 3A,B*; *Schirenbeck et al., 2005*). Formin activity is not required for cortical recruitment of DdMyo7 as it is found to localize normally to the cortex of *dDia2* null (*Figure 3C,D*). These data reveal that DdVASP or its actin polymerization activity is critical for DdMyo7 recruitment and suggests that this bundler/polymerase acts upstream of DdMyo7.

## Selective recruitment of DdMyo7 requires specific actin polymerization factors

The finding that DdVASP is critical for DdMyo7 cortical targeting raises the question of how it is acting to recruit this myosin. One simple explanation is that DdMyo7 is targeting to regions of the cell cortex where dynamic actin polymerization occurs. This hypothesis was tested by examining if robust recruitment of DdMyo7 could occur in the absence of DdVASP when actin polymerization was stimulated using different manipulations. Treatment of *Dictyostelium* with latrunculinA treatment followed by washout produces robust actin waves (*Gerisch et al., 2004*). These waves are made of a dense Arp2/3 branched actin meshwork and class I myosins (MyoB, MyoC) (*Jasnin et al., 2019*; *Brzeska et al., 2020*). Control or *vasp* null cells expressing both RFP-LimEΔcoil (a marker for F-actin) and GFP-DdMyo7 were induced to form waves that are readily apparent as broad circles of actin that emanate outwards toward the cell periphery. Robust actin waves marked by RFP-LimEΔcoil were observed but DdMyo7 was not observed in the waves in either cell line (n = 6 per genotype; *Figure 4A*, *Figure 4—figure supplement 1A,B*). Next, *vasp*-cells were treated with jasplakinolide to stimulate actin polymerization (*Figure 4—figure supplement 1C*). No increase in cortical targeting of DdMyo7 was observed in *vasp* nulls treated with jasplakinolide when compared to untreated cells (*Figure 4B,E*). Blocking capping protein by overexpressing the capping protein inhibitor V-1 stimulates filopodia formation in *Dictyostelium* (*Figure 4—figure supplement 1D*, *Jung et al., 2016*). V-1 also stimulates cortical actin network formation in *vasp* null *Dictyostelium* (*Figure 4C*). However, no

**Table 1.** Cortical recruitment ratio of DdMyo7 and filopodia per cell for GFP-DdMyo7/*myo7* null cells treated with various pharmacological compounds.

| Controls | Buffer only | 1% DMSO | Noco. Ctrl. |
|---|---|---|---|
| Percent of cells with filopodia | 42 | 38 | 29 |
| Filopodia number + SEM | 2.01 ± 0.11 | 1.97 ± 0.09 | 1.5 ± 0.17 |
| Cortex:cytoplasm ratio + SEM | 1.18 ± 0.01 | 1.2 ± 0.02 | 1.31 ± 0.03 |
| N, n | 4, 238 | 3, 229 | 4138 |
| **CytochalasinA** | **1 μM** | **5 μM** | **30 μM** |
| Percent of cells with filopodia | 0 | 8.5 | 19 |
| Filopodia number + SEM | 0 | 1.3 ± 0.04 | 0.1 ± 0.06 |
| Cortex:cytoplasm ratio + SEM | 1.02 ± 0.02 | 1.07 ± 0.01 | 1.04 ± 0.02 |
| N, n | 3, 66 | 3, 234 | 1, 42 |
| **LatrunculinA** | **1 μM** | **5 μM** | **15 μM** |
| Percent of cells with filopodia | 19 | 4.7 | 20 |
| Filopodia number + SEM | 1.73 ± 0.06 | 1.33 ± 0.05 | 0.24 ± 0.06 |
| cortex:cytoplasm ratio + SEM | 1.12 ± 0.01 | 1.02 ± 0.01 | 1.08 ± 0.02 |
| N, n | 4, 387 | 3, 127 | 1, 74 |
| **Jasplakinolide** | **15 nM** | **50 nM** | **100 nM** |
| Percent of cells with filopodia | 55 | 72 | 48 |
| Filopodia number + SEM | 1.47 ± 0.23 | 3.89 ± 0.2 | 2.21 ± 0.09 |
| Cortex:cytoplasm ratio + SEM | 1.19 ± 0.03 | 1.34 ± 0.03 | 1.24 ± 0.02 |
| N, n | 1, 84 | 3, 219 | 3, 337 |
| **LY294002/wortmannin** | **20 μM LY294002** | **60 μM LY294002** | **2 μM WM** |
| Percent of cells with filopodia | 22 | 40 | 30 |
| Filopodia number + SEM | 0.56 ± 0.44 | 0.87 ± 0.12 | 0.84 ± 0.16 |
| Cortex:cytoplasm ratio + SEM | 1.22 ± 0.08 | 1.17 ± 0.04 | 1.16 ± 0.02 |
| N, n | 2, 9 | 4, 194 | 2, 101 |
| **CK666** | **25 μM** | | |
| Percent of cells with filopodia | 10.7 | | |
| Filopodia number + SEM | 0.145 ± 0.03 | | |
| Cortex:cytoplasm ratio + SEM | 1.12 ± 0.01 | | |
| N, n | 3, 366 | | |
| **Nocodazole** | **5 μM** | **15 μM** | **50 μM** |
| Percent of cells with filopodia | 32 | 14 | 8 |
| Filopodia number + SEM | 1.74 ± 0.22 | 1.27 ± 0.11 | 1.1 ± 0.09 |
| Cortex:cytoplasm ratio + SEM | 1.15 ± 0.03 | 1.12 ± 0.01 | 1.1 ± 0.01 |
| N, n | 2, 59 | 4, 133 | 2, 127 |

Percent of all cells with at least one filopodia. Average number of filopodia per cells from cells with at least one filopodia. Cortex:cytoplasm ratio is intensity ratio of a 0.8 μm band around the periphery compared to the cytoplasm. N is number of experiments, n is number of cells. SEM is standard error of the mean. See also **Table 1—source data 1–4**.

The online version of this article includes the following source data for Table 1:

**Source data 1.** Filopodia per cell values for each cell for lines analyzed in **Table 1**.

**Source data 2.** Statistical test results for filopodia per cell data in **Table 1** (see also **Figure 2—figure supplement 1**).

**Source data 3.** Cortex: cell ratio values for each cell for lines presented in **Table 1**.

**Source data 4.** Statistical test results for cortex: cell ratio data in **Table 1** (see also **Figure 2—figure supplement 1**).

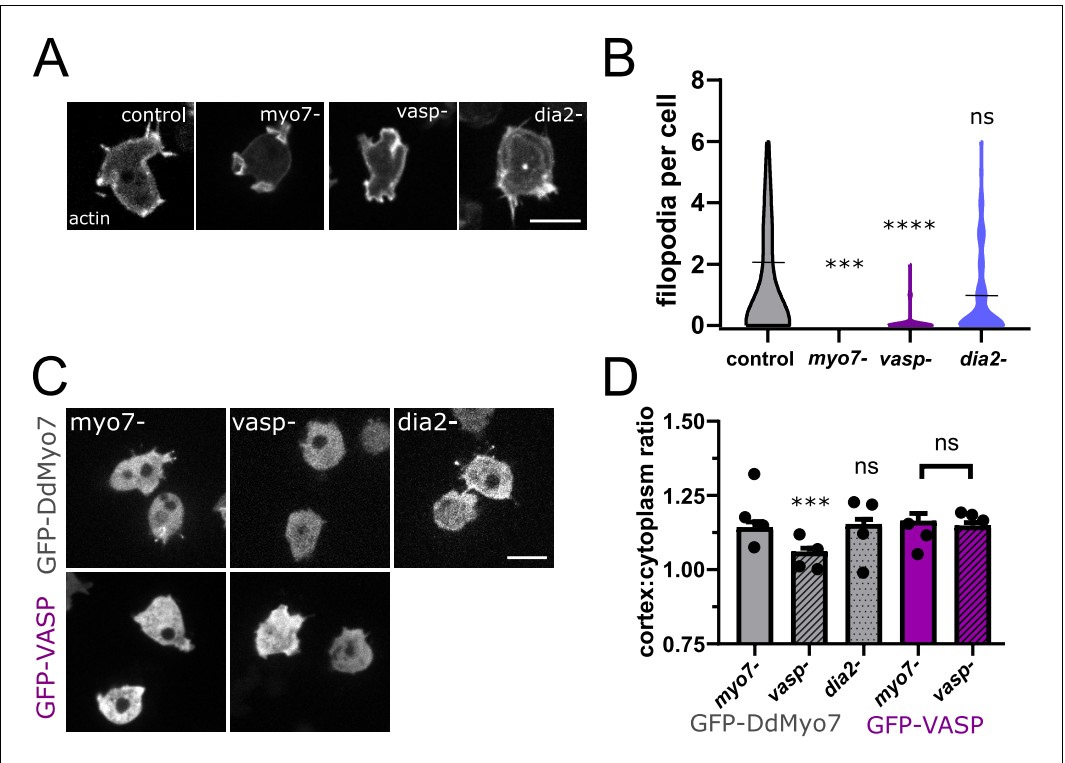

**Figure 3.** VASP is required for DdMyo7 cortical recruitment. (**A**) Confocal images of wild type, *myo7* null, *vasp* null or *dia2* null cells expressing RFP-Lifeact (actin). (**B**) Violin plot of number of filopodia per cell (see also **Figure 3— source data 1**). (**C**) Micrographs of cells expressing GFP-DdMyo7 (top) or GFP-VASP in *myo7* null, *vasp* null or *dDia2* null cells. (**D**) Quantification of the cortical band (0.8 μm of periphery) relative to the cytoplasmic intensity of either GFP-Myo7 or GFP-VASP. (**A,C**) Scale bar is 10 μm (see also **Figure 3—source data 3**). (**B, D**) One-way ANOVA with multiple comparison correction or student's t-test to compare GFP-VASP, ns, not significant, p***<0.001, p****<0.0001, circles are experimental means (see also **Figure 3—source data 2** and **4**).
The online version of this article includes the following source data and figure supplement(s) for figure 3:

**Source data 1.** Number of filopodia per cell counted for each cell in strains in **Figure 3B**.
**Source data 2.** Statistical test results for **Figure 3B**.
**Source data 3.** Cortex: cell ratio values for each cell for lines analyzed in **Figure 3D**.
**Source data 4.** Statistical test results for **Figure 3D**.
**Figure supplement 1.** VASP is not present in DdMyo7 immunoprecipitates.

increase in targeting was observed in *vasp* nulls with induced overexpression of V-1 (**Figure 4B,D**), nor was filopodia formation restored (**Figure 4F**).

Activation of formins can also result in robust polymerization of actin filaments that are organized into linear bundles (**Breitsprecher and Goode, 2013**). These polymerases are tightly controlled by autoinhibition with the conserved DAD domain binding to the N-terminal DID domain, blocking actin nucleation activity. Mutation of conserved basic residues in the C-terminal DAD domain creates a constitutively activated (CA) formin (**Wallar et al., 2006**). A pair of charged residues are present in the dDia2 DAD domain and these were mutated - R1035A, R1036A - to create a CA formin (see alignment in **Figure 4—figure supplement 1E**). Either dDia2 or dDia2-CA overexpressing *vasp* null cells exhibit restored and increased DdMyo7 cortical targeting (**Figure 4B,E**). Overexpression of dDia2-CA also promoted a modest rescue of filopodia formation (**Figure 4F**; **Table 2**). Linescans of cells stained with phalloidin show that while *vasp* nulls have less cortical F-actin than wild type cells, expression of dDia2-CA in *vasp* nulls resulted in increased cortical F-actin (**Figure 4D**). In summary, stimulation of actin polymerization by inducing actin waves, treating cells with jasplakinolide, or blocking capping protein is not sufficient to promote DdMyo7 cortical recruitment in *vasp* null cells. In contrast, expression of activated formin is sufficient for cortical recruitment in *vasp* null cells. This

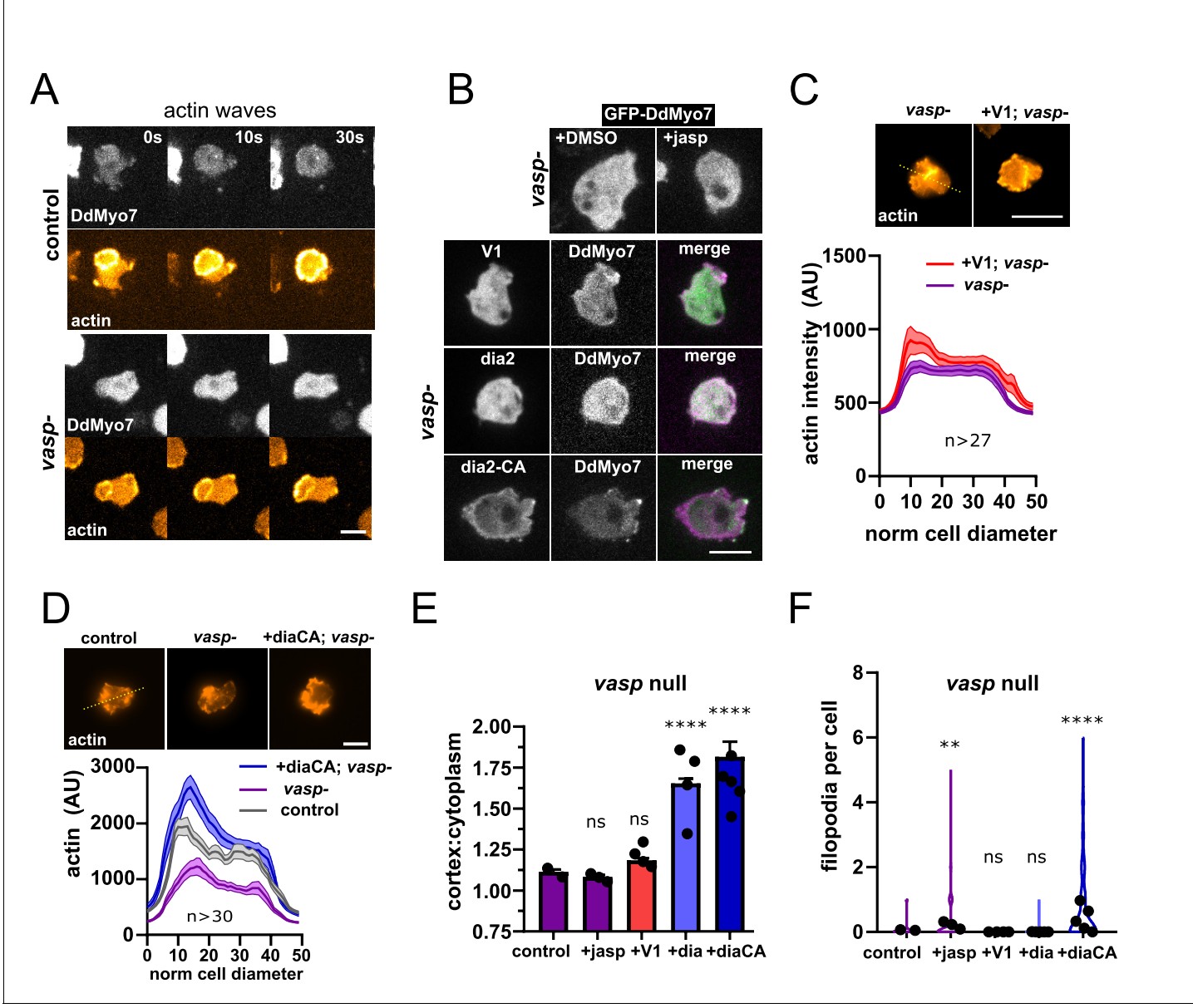

**Figure 4.** Linear actin polymerization drives DdMyo7 to the cortex. (A) Image series showing DdMyo7 is absent from latrunculinA-induced actin waves in control (top) or *vasp null* (bottom) cells. (B) (top) Confocal images of GFP-DdMyo7 in *vasp* null cells treated with either DMSO or 50 nM jasp treatment. (bottom) Images of *vasp* null cells expressing GFP or mCherry DdMyo7 and different actin modulating proteins (GFP-V1, green; GFP-dia2WT, green; RFP-dia2CA, magenta). (C, D) Average actin intensity (phalloidin staining, top) of cells through the longest cell axis. The line is the mean and the shaded area is the SEM (graphs, bottom) (see also *Figure 4—source data 1*). (A–D), Scale bar is 10 µm. (E) Quantification of the cortical band intensity of DdMyo7 in *vasp* null cells, with no treatment, treated with jasp, or also overexpressing V-1, dia2, or dia2-CA (see also *Figure 4—source data 2*). (F) Violin plot of filopodia per cell (see also *Figure 4—source data 4*). (E-F). One-way ANOVA with multiple comparison correction, ns, not significant, **$p<0.05$, p****$<0.0001$ (see also *Figure 4—source data 3* and *5*).

The online version of this article includes the following source data and figure supplement(s) for figure 4:

**Source data 1.** Actin line scans for *Figure 4C,D*.
**Source data 2.** Cortex: cell ratio values for each cell for lines analyzed in *Figure 4E*.
**Source data 3.** Statistical test results for *Figure 4E*.
**Source data 4.** Filopodia per cell values for each cell for lines analyzed in *Figure 4F*.
**Source data 5.** Statistical test results for *Figure 4F*.
**Figure supplement 1.** Effects of actin modulating drugs and proteins.
**Figure supplement 1—source data 1.** Filopodia per cell values for lines shown in *Figure 4—figure supplement 1D*.
**Figure supplement 1—source data 2.** Statistical test results for *Figure 4—figure supplement 1D*.

**Table 2.** Quantification of filopodia number and cortical targeting.

| | | DdMyo7 | KKAA | E386V | E386V-KKAA | I426A | CAAX | Tail | GFP-VASP | | | |
|---|---|---|---|---|---|---|---|---|---|---|---|---|
| *myo7-* | percent of cells with filopodia | 54 | 80 | 1 | 3 | 22 | 58 | n.c. | n.c. | | | |
| | filopodia number + SEM | 2.53 ± 0.15 | 5.32 ± 0.33 | 1 ± 0 | 1.5 ± 0.09 | 1.75 ± 0.15 | 2.87 ± 0.14 | n.c. | n.c. | | | |
| | cortex:cytoplasm ratio + SEM | 1.18 ± 0.02 | 1.67 ± 0.06 | 1 ± 0.01 | 1.32 ± 0.04 | 1.19 ± 0.04 | 1.39 ± 0.03 | 1.61 ± 0.07 | 1.18 ± 0.29 | | | |
| | N, n | 4, 198 | 3, 133 | 3, 186 | 3, 59 | 3, 37 | 3, 237 | 3, 55 | 3, 91 | | | |
| | | GFP-DdMyo7 | VASP-ΔTET | VASP -WT | VASP-FAB-K-E | GFP-VASP | DdMyo7-CAAX | DdMyo7-Tail | DdMyo7-KKAA | GFP-Dia2 | mCherry-Dia2-CA | GFP-V1 |
| *vasp-* | percent of cells with filopodia | 7 | 35 | 69 | 14 | n.c. | 4 | n.c. | 4 | 0 | 29 | 0 |
| | filopodia number + SEM | 1.25 ± 0.07 | 3.78 ± 0.31 | 3.07 ± 0.25 | 1.39 ± 0.03 | n.c. | 1.2 ± 0.03 | n.c. | 1 ± 0 | 0 ± 0 | 2.03 ± 0.24 | 0 ± 0 |
| | cortex:cytoplasm ratio + SEM | 1.06 ± 0.01 | 1.2 ± 0.02 | 1.47 ± 0.05 | 1.19 ± 0.01 | 1.18 ± 0.33 | 1.24 ± 0.01 | 1.35 ± 0.02 | 1.29 ± 0.02 | 1.65 ± 0.03 | 1.82 ± 0.09 | 1.12 ± 0.02 |
| | N, n | 3, 118 | 3, 131 | 3, 81 | 3, 322 | 3, 200 | 4, 239 | 3, 213 | 3, 193 | 5447 | 8,94 | 3,35 |
| | | GFP-DdMyo7 | GFP-DdMyo7-CAAX | GFP-DdMyo7-Tail | GFP-DdMyo7-KKAA | GFP-VASP | | | | | | |
| Ax2 control | percent of cells with filopodia | 45 | 54 | n.c. | 73 | n.c. | | | | | | |
| | filopodia number + SEM | 2.36 ± 0.2 | 2.9 ± 0.15 | n.c. | 3.65 ± 0.2 | n.c. | | | | | | |
| | cortex:cytoplasm ratio + SEM | 1.35 ± 0.03 | 1.45 ± 0.04 | 1.41 ± 0.02 | 1.79 ± 0.07 | 1.16 ± 0.25 | | | | | | |
| | N, n | 4, 124 | 5, 266 | 4, 351 | 3, 219 | 4, 216 | | | | | | |

The percent of all cells with at least one filopodia. Average number of filopodia per cells from cells with at least one filopodia. Cortex:cytoplasm ratio is intensity ratio of a 0.8 µm band around the periphery compared to the cytoplasm. N is number of experiments, n is number of cells. SEM is standard error of the mean. GFP-VASP and GFP-DdMyo7-Tail fail to efficiently target to filopodia tips and thus were not counted (n.c.) in this analysis. (See source data: *Figure 4—source data 1* and *2*; *Figure 5—source data 1–3*; *Figure 6—source data 1–5*; *Figure 7—source data 1*, *2*, *4* and *5*).

strongly suggests that DdMyo7 cortical targeting specifically requires the activity of actin regulators that generate parallel actin filaments.

## Mechanism of VASP-mediated actin polymerization required for DdMyo7 recruitment

The properties of VASP critical for DdMyo7 recruitment were investigated using separation of function mutations. VASP accelerates the rate of actin polymerization, bundles actin filaments and blocks capping protein from binding to the barbed ends of actin filaments (*Breitsprecher et al., 2008*; *Breitsprecher et al., 2011*; *Hansen and Mullins, 2010*). VASP is a constitutive tetramer that can bundle actin by binding to the sides of filaments (see *Figure 5A*; (*Bachmann et al., 1999*; *Breitsprecher et al., 2008*; *Brühmann et al., 2017*). DdVASP F-actin binding is mediated by a region within the EVH2 domain (aa 264–289) and tetramerization by a region at the C-terminus (aa 341–375) (*Schirenbeck et al., 2006*; *Breitsprecher et al., 2008*). Two different mutants were used to test if the VASP bundling activity is required for DdMyo7 recruitment (*Figure 5D*). A monomeric VASP was created by deleting the tetramerization domain (Δtet; Δ341–375)) and an F-actin binding mutant was generated in which conserved lysines were mutated to glutamate (FAB K-E; KR275, 276EE + K278E + K280E; based on *Hansen and Mullins, 2010*). The FAB K-E mutations are predicted to slow but not eliminate actin polymerization (*Breitsprecher et al., 2008*; *Schirenbeck et al., 2006*; *Applewhite et al., 2007*; *Hansen and Mullins, 2010*). Co-expression of either DdVASP-Δtet or DdVASP-FAB K-E with DdMyo7 in *vasp* null cells cannot fully restore DdMyo7 cortical recruitment (*Figure 5B,E*). The monomeric VASP-Δtet mutant partially rescues filopodia

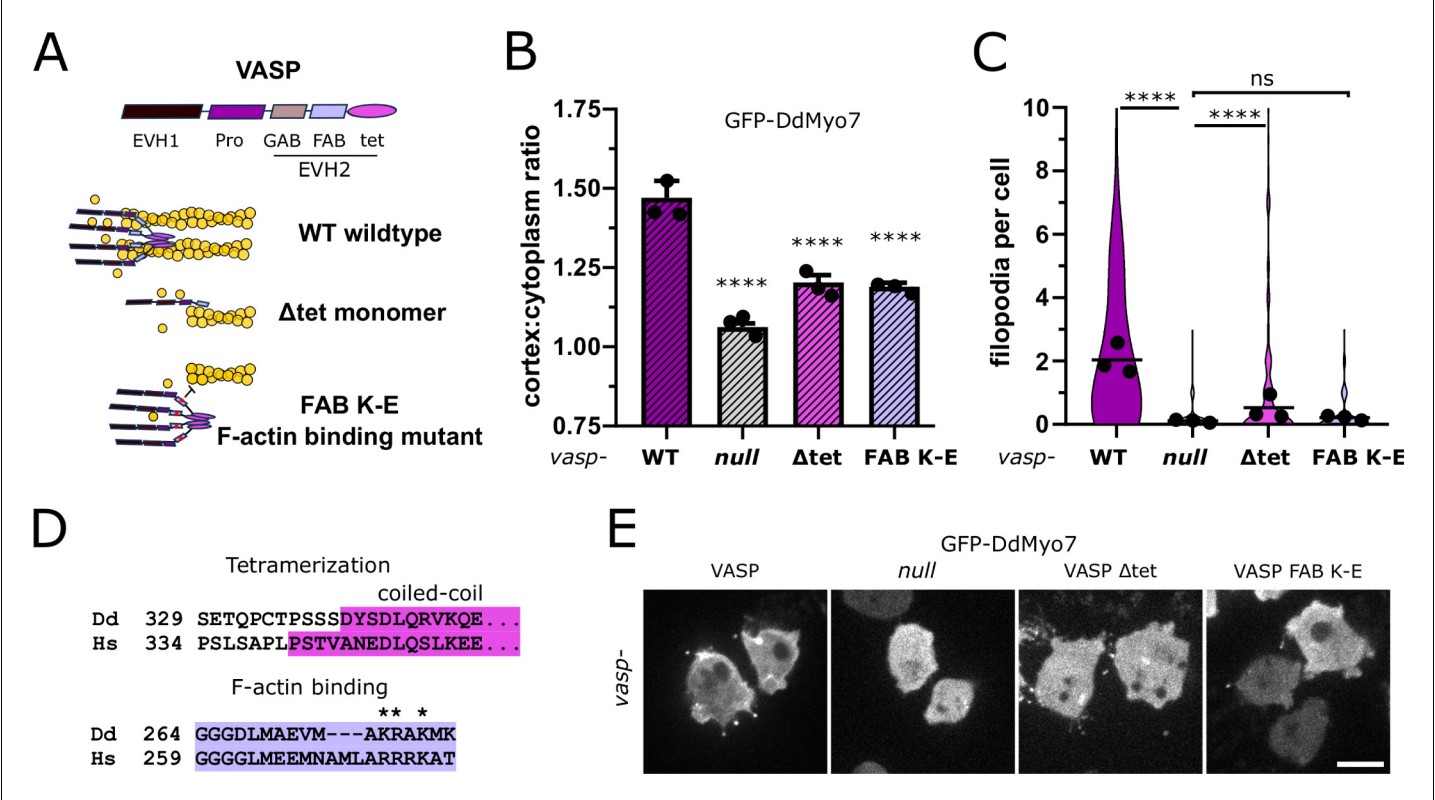

**Figure 5.** Reduced cortical recruitment of DdMyo7 by VASP mutants. (**A**) Schematic of domains of DdVASP (top) and proposed interaction of DdVASP wildtype, monomeric (Δtet), and F-actin binding (FAB K-E) mutant with actin filaments. (**B**) Quantification of the cortical recruitment of GFP-DdMyo7 co-expressed in the *vasp* null with wildtype or mutant DdVASP (non-fluorescent) rescue constructs (see also *Figure 5—source data 1*). (**C**) Quantification of GFP-DdMyo7 positive filopodia per cell of *vasp* null cells with wildtype or mutant DdVASP rescue constructs (see also *Figure 5—source data 3*). (**B–C**) Circles represent experimental means. One-way ANOVA with multiple comparison correction, p****<0.0001, ns not significant (see also *Figure 5—source data 2* and *4*). (**D**) Clustal Omega alignment of *Dictyostelium* and human VASP sequences with conserved domains highlighted and mutated residues starred. (**E**) Micrographs of GFP-DdMyo7 in *vasp* nulls, or *vasp* nulls expressing wildtype DdVASP or mutant DdVASP rescue constructs. Scale bar is 10 μm.

The online version of this article includes the following source data for figure 5:

**Source data 1.** Cortex: cell ratio values for each cell for lines analyzed in *Figure 5B*.

**Source data 2.** Statistical test results for *Figure 5B*.

**Source data 3.** Filopodia per cell values for each cell for lines analyzed in *Figure 5C*.

**Source data 4.** Statistical test results for *Figure 5C*.

formation while the FAB K-E mutant does not (*Figure 5C*). The residual filopodia forming activity of the VASP-Δtet monomer could be attributed to its remaining anti-capping activity that is lost in the F-actin binding mutant (*Breitsprecher et al., 2008*; *Hansen and Mullins, 2010*). Both mutants are predicted to lack any bundling activity and show decreased recruitment of DdMyo7 to the cortex. These data indicate that bundling linear F-actin at the membrane could promote DdMyo7 targeting.

## Release of DdMyo7 autoinhibition dictates the spatial restriction of its recruitment

DdMyo7 is enriched at the leading edge of cells, in contrast to the tail alone that localizes all around the cortex (*Figure 1F,I*). This difference suggests that head-tail autoinhibition restricts DdMyo7 localization. DdVASP is required for cortical localization but the apparent lack of interaction between DdMyo7 and DdVASP suggests an indirect role. Furthermore, the GFP-DdMyo7 tail domain in *vasp* null cells targets to the cortex efficiently (*Figure 6A–C*). This confirms that direct DdMyo7 tail - DdVASP binding is not required for localization to the cortex. Together these results suggested that

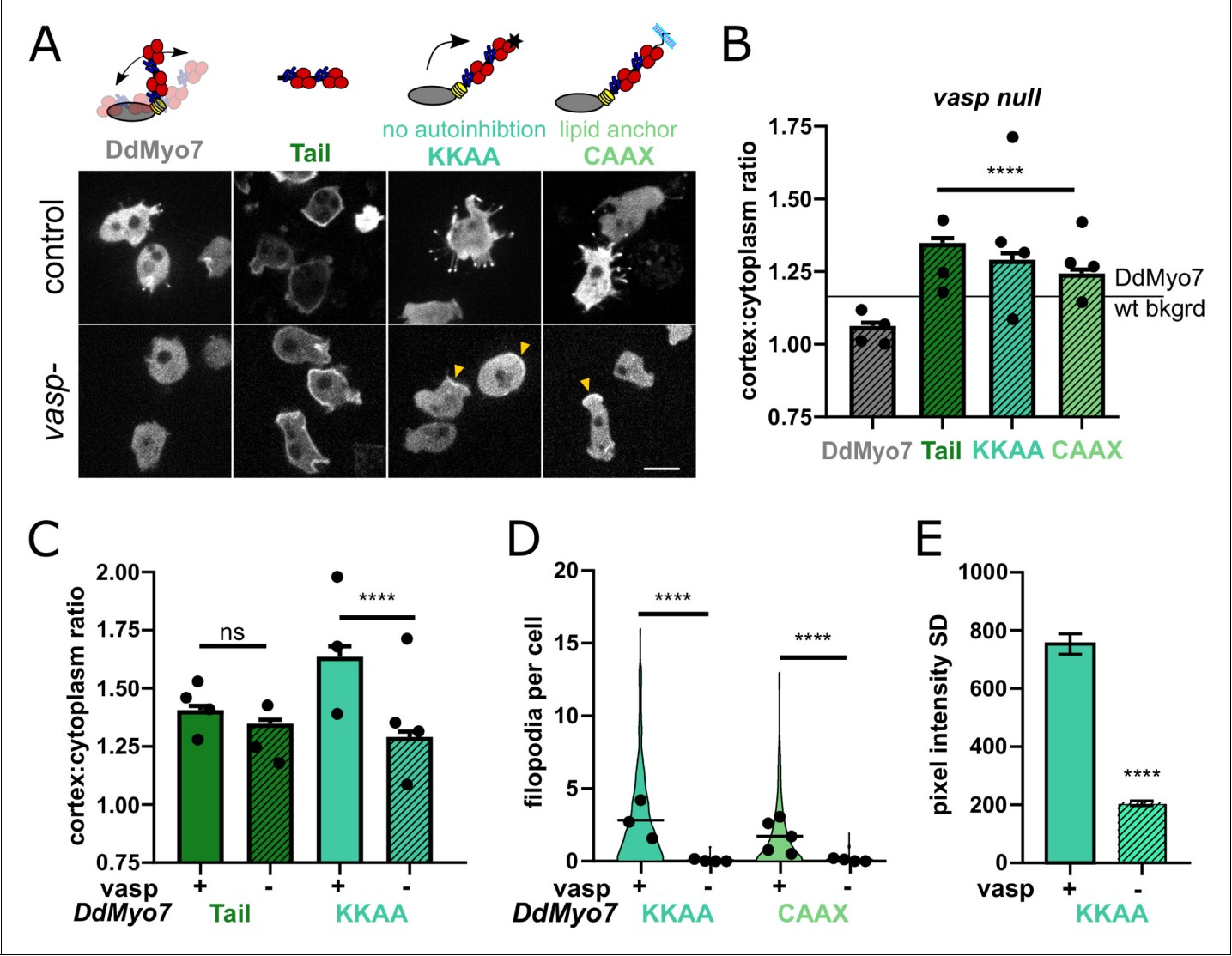

**Figure 6.** VASP-mediated actin assembly relieves DdMyo7 head-tail autoinhibition to promote targeting and filopodia formation. (A) (top) Diagrams depicting mutants analyzed. (bottom) Micrographs of GFP-DdMyo7 fusion proteins in control and *vasp* null cells, scale bar is 10 μm. Arrows indicate cortical enrichment of DdMyo7. (B) Quantification of cortical recruitment of GFP-DdMyo7 and variants in *vasp* null cells (see also *Table 2* and *Figure 6—source data 1*). The line represents the mean GFP-DdMyo7 recruitment in wild type cells. (C) Comparison of cortical targeting of activated DdMyo7-KKAA or tail in *vasp* null versus control cells (see also *Figure 6—source data 1* and *2*). (D) Quantification of number of filopodia per cell in control or *vasp* null cells (see also *Table 2*; *Figure 6—source data 3*). (B–D) Circles represent experimental means. One way ANOVA with multiple comparison test, ns not significant, p***<0.001, p****<0.0001, ns, not significant (see also *Figure 6—source data 2* and *4*). (E) Quantification of the cortical band intensity variation of DdMyo7-KKAA in control versus *vasp* null cells (see also *Figure 6—source data 5*). Students t-test ****p<0.0001. The online version of this article includes the following source data for figure 6:

**Source data 1.** Cortex: cell ratio values for each cell for lines analyzed in *Figure 6B*.

**Source data 2.** Statistical test results for *Figure 6B,C*.

**Source data 3.** Filopodia per cell values for each cell for lines analyzed in *Figure 6D*.

**Source data 4.** Statistical test results for *Figure 6D*.

**Source data 5.** Values of the cortical standard deviation measurements (cortical asymmetry) measured for each cell for *Figure 6E*.

VASP mediated actin networks release autoinhibition of DdMyo7. If so, then loss of autoinhibition regulation of DdMyo7 should eliminate the requirement of VASP for DdMyo7 to target to the cortex. To test this, two highly conserved charged residues at the extreme C-terminus of DdMyo7 (K2333, K2336) essential for head-tail autoinhibition were mutated (*Yang et al., 2009*;

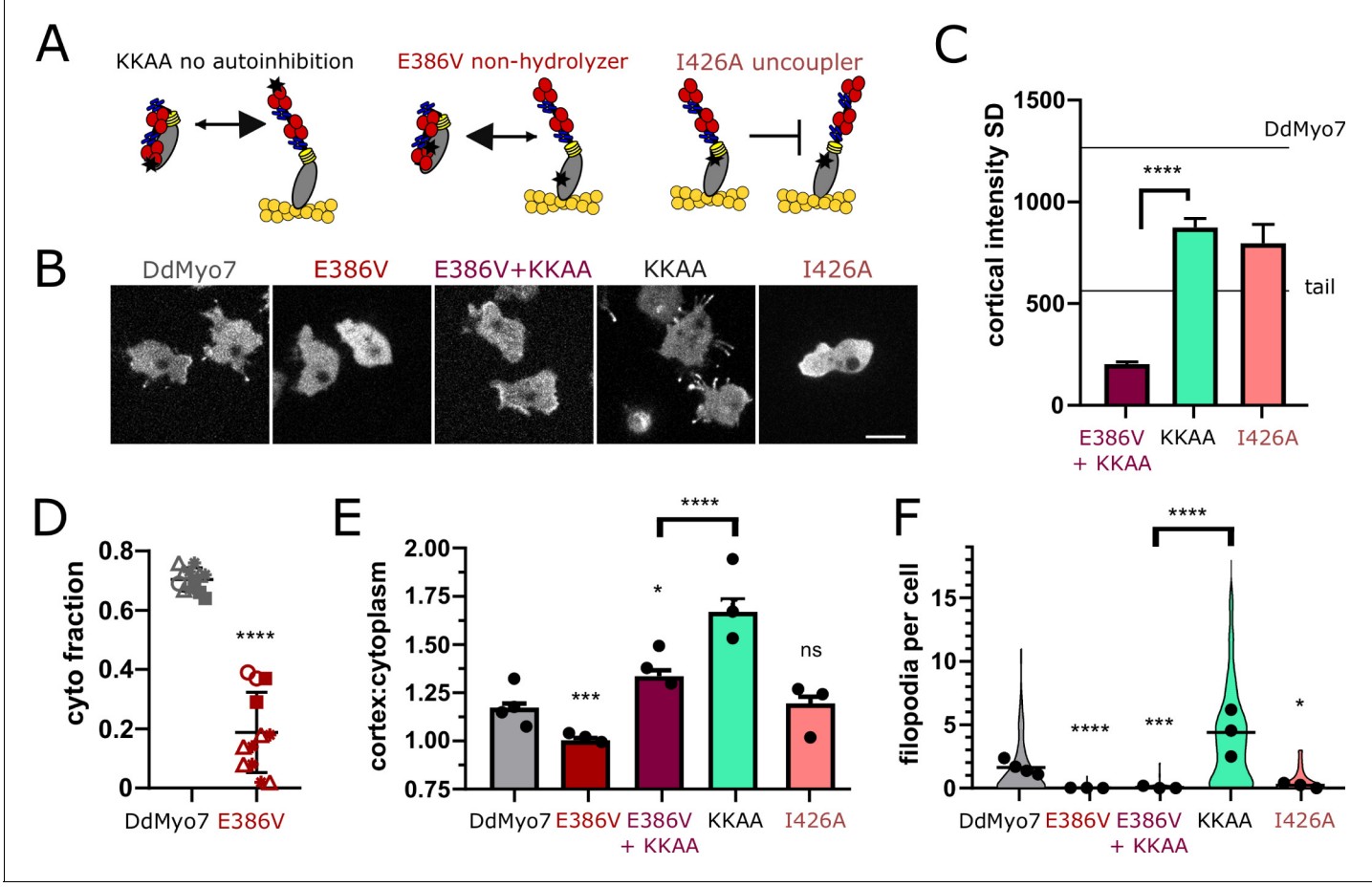

**Figure 7.** DdMyo7 motor activity is required to release autoinhibition. (**A**) Schematic of proposed effect of mutations on DdMyo7 function (see alignment in *Figure 7—figure supplement 1*). (**B**) Confocal images of *myo7* null cells expressing GFP-DdMyo7 fusion proteins, scale bar is 10 μm. (**C**) Quantification of the cortical band intensity variation. Mean lines from *Figure 1I* data on graph for comparison. DdMyo7 versus I426A uncoupler, p=0.07, not significant (see also *Figure 7—source data 1* and *2*). (**D**) Fraction of DdMyo7 cosedimenting with the cytoskeleton, symbols with the same shape are technical replicates, students t-test ****p<0.0001 (see also *Figure 7—figure supplement 1*, *Figure 7—source data 3*). (**E**) Quantification of cortical recruitment of DdMyo7 and mutants (see also *Figure 7—source data 4*). (**F**) Filopodia number per cell of wildtype and DdMyo7 mutants (see also *Figure 7—source data 6*). (**E–F**) Data for KKAA is taken from *Figure 6*, experimental means shown as circles. (**C, E, and F**) One-way ANOVA with multiple comparison correction, p*<0.05, p***<0.001, p****<0.0001, ns not significant (see also *Figure 7—source data 5* and *7*).

The online version of this article includes the following source data and figure supplement(s) for figure 7:

**Source data 1.** Values of the cortical standard deviation measurements (cortical asymmetry) measured for each cell for *Figure 7C*, see also *Figure 1—source data 1* and *Figure 6—source data 5*.
**Source data 2.** Statistical test results for *Figure 7C*.
**Source data 3.** Data and statistics test results for *Figure 7D*.
**Source data 4.** Cortex: cell ratio values for each cell for lines analyzed in *Figure 7E*.
**Source data 5.** Statistical test results for *Figure 7E*.
**Source data 6.** Filopodia per cell values for each cell for lines analyzed in *Figure 7F*.
**Source data 7.** Statistical test results for *Figure 7F*.
**Figure supplement 1.** Conservation of the DdMyo7 motor domain.

*Petersen et al., 2016*). Overexpression of this constitutively active mutant (DdMyo7-KKAA) stimulates filopodia formation and increases cortical localization in wild-type cells (*Table 2*, *Petersen et al., 2016*; *Arthur et al., 2019*). DdMyo7-KKAA was expressed in *vasp* null cells where it targets to the cortex (*Figure 6A–B*). The cortical targeting of DdMyo7-KKAA in *vasp* nulls is not as robust as seen in control cells (*Figure 6C*), and it is localized more uniformly around the cortex in *vasp* nulls (less enriched in the leading edge or pseudopod) compared to controls (*Figure 6E*). These

observations are consistent with VASP-mediated actin assembly having a role in local activation and recruitment of DdMyo7 to the cortex.

## VASP and cortical DdMyo7 are both required for filopodia formation

The finding that loss of head-tail autoinhibition restores cortical recruitment of DdMyo7 raised the question of whether an autoactivated motor is sufficient to rescue the filopodia formation defect seen in the *vasp* null cells. Expression of DdMyo7-KKAA in *vasp* nulls did not rescue their filopodia formation defect (*Figure 6D*), indicating that a specific DdVASP activity is required for filopodia initiation. The requirement for both cortical DdVASP and DdMyo7 for filopodia formation was further tested by targeting of the myosin to the cortex or membrane by fusing a prenylation sequence to its C-terminus (DdMyo7-CAAX, adapted from *Weeks et al., 1987*). DdMyo7-CAAX was robustly localized to the cortex in *myo7* null, wild type and *vasp* null cells (*Figure 6A,B*; *Table 2*). Expression of DdMyo7-CAAX in wildtype or *myo7* nulls cells significantly stimulated filopodia formation (*Figure 6A,D*, *Table 2*). However, filopodia are not formed when DdMyo7-CAAX was expressed in *vasp* null cells (*Figure 6A,D*). These results establish that targeting DdMyo7 to the membrane alone is not sufficient for filopodia initiation and that DdVASP must also be present.

## Role of myosin motor activity in targeting and filopodia formation

The data suggest that activation of autoinhibited DdMyo7 is promoted at the front of the cell in regions rich in newly polymerized F-actin. Actin binding by myosin involves conformational changes in the myosin head (*Houdusse and Sweeney, 2016*) and could destabilize the interactions between the motor head and the tail. To test this, two DdMyo7 mutants were designed, each with mutations in highly conserved connectors of the motor domain. Their functions are established by studies performed on the highly conserved motor domain of DdMyo2 (*Sasaki et al., 2003*; *Friedman et al., 1998*). The non-hydrolyzer mutant (E386V) binds ATP but cannot hydrolyze it, and thus stays in weak actin-binding state, while the uncoupler mutant (I426A) can undergo conformational changes allowing strong interactions with F-actin but these conformational changes are not transmitted to the lever arm to produce force (illustrated in *Figure 7A*).

The non-hydrolyzer (E386V) has a mutated glutamate in Switch II that is required to assist hydrolysis (*Figure 7—figure supplement 1A*, *Friedman et al., 1998*). The non-hydrolyzer failed to efficiently cosediment with the actin cytoskeleton upon centrifugation (*Figure 7D*, *Figure 7—figure supplement 1B*) indicating that it was indeed a weak actin binding mutant. The DdMyo7-E386V mutant does not target efficiently to the cell periphery (*Figure 7B,E*) or rescue filopodia formation (*Figure 7F*; *Table 2*). The lack of cortical localization by the non-hydrolyzer suggests that the tail remains bound to the motor, favoring an autoinhibited conformation, and is not free to bind to the cortex (see *Figure 7A*). To test this, the autoinhibition point mutations (KKAA) were introduced into the non-hydrolyzer to disrupt the head/tail interface. Blocking autoinhibition (KKAA) in the non-hydrolysis mutant (DdMyo7-E386A + KKAA) strikingly restores cortical targeting (*Figure 7B,E*; *Table 2*) but not filopodia formation (*Figure 7F*; *Table 2*).

A second motor mutation, I426A, resides in the relay helix and disrupts the interface with the converter that is critical to direct the swing of the lever arm during force generation. This mutant uncouples ATP hydrolysis from force generation (*Figure 7—figure supplement 1A*, *Sasaki et al., 2003*). Thus, the uncoupler (DdMyo7-I426A) undergoes actin-activated ATP hydrolysis, likely due to conformational changes directed from binding to actin, but it cannot exert force (illustrated in *Figure 7A*). DdMyo7-I426A cortical targeting is similar to what is seen for wild-type DdMyo7 (*Figure 7B,E*). This indicates that actin binding destabilizes the autoinhibited form in this mutant. DdMyo7-I426A fails to efficiently rescue the filopodia formation defect of *myo7* null cells despite its targeting to the cortex (*Figure 7F* and *Table 2*), establishing that force generation by DdMyo7 is essential for filopodia initiation.

The uncoupler mutant motor is predicted to have normal actin binding and indeed its cortical asymmetry is similar to wild-type DdMyo7 (*Figure 7B,C*). In contrast, the motor of the non-hydrolyzer DdMyo7-E386A + KKAA mutant has weak actin binding so interaction with the cortex would be directed by the tail. As predicted, the DdMyo7-E386A + KKAA mutant is localized more uniformly around the cortex (low cortical SD, *Figure 7B,C*). These observations provide support for the

model that while the tail aids overall cortical localization, motor-actin binding refines leading edge targeting of DdMyo7.

## Discussion

Filopodia formation in amoeba requires both an MF myosin and VASP (*Tuxworth et al., 2001*; *Han et al., 2002*). The motor domain of the MF myosin DdMyo7, not the tail, targets it to actin-rich dynamic regions of the cortex where filopodia emerge (*Figure 1*). Actin network dynamics generated by DdVASP activity in vivo are important for myosin localization, promoting release of head-tail autoinhibition (*Figure 6*), freeing and activating the motor domain for binding to actin at the leading edge (*Figures 1* and *7*). This controlled activation of MF myosin leads to tuned activity at the right time and the right place to concentrate motors and facilitate filopodia formation, mainly from the leading edge of the cell. The role of DdVASP goes beyond simple recruitment of the motor since targeting of DdMyo7 at the membrane is not sufficient for filopodia initiation in *vasp* null cells (*Figure 6A,D*). Thus, both MF myosin and DdVASP together are needed to reorganize and polymerize actin at the cell cortex for filopodia to form (*Figures 3* and *4*). The data presented here support a model that DdMyo7 is mostly in a closed confirmation or OFF state (*Figure 8A*). VASP activity, possibly bundling or polymerizing unbranched actin at the leading edge is upstream of DdMyo7 activation (*Figure 8B*). DdMyo7 is activated by motor binding to a DdVASP generated actin network (*Figure 8C*), specifically the open state is stabilized by motor binding to the linear, bundles of actin resulting from DdVASP activity. Together, DdVASP and DdMyo7 drive filopodia initiation by further bundling and polymerizing actin (*Figure 8D*).

### The role of DdVASP in myosin recruitment

Disruption of the dynamic actin network at the cortex that occurs by loss of *vasp* (*Figure 4D*) or by treatment with anti-actin drugs such as cytoA or latA results in loss of cortical localization of DdMyo7 (*Figure 2G,H*). *Dictyostelium* VASP is a potent actin polymerase, its activity speeds actin elongation, bundles filaments and blocks capping proteins from binding to the growing ends of actin filaments (*Breitsprecher et al., 2008*). Both its actin polymerization and bundling activities could be important for DdMyo7 recruitment (*Figure 5*). In the absence of DdVASP, increased cortical actin generated by the formin dDia2 can also recruit and activate DdMyo7 (*Figure 4B-E*). However, recruitment is not restored by several other means of increasing actin polymerization, including generating actin waves, polymerizing actin with jasplakinolide or blocking capping protein by overexpression of V-1. Thus, DdVASP does not simply recruit DdMyo7 by generating new actin polymers, but rather it is the nature of the network that DdVASP builds, likely growing linear actin filaments that are bundled together in parallel, that is critical for activation of DdMyo7.

The actin cortex of *vasp* nulls is less dense than found for wild-type cells as revealed by phalloidin staining, suggesting that a robust leading edge actin network collapses without VASP (*Figure 4D*).

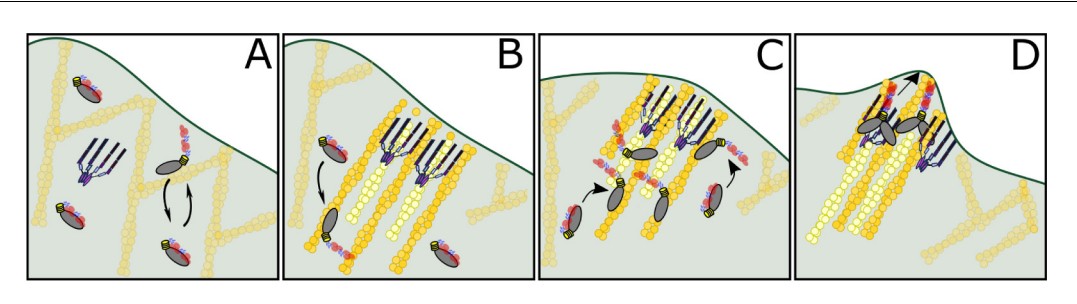

**Figure 8.** Model of DdMyo7 and VASP mediated filopodia initiation. (**A**) The leading edge of the cell has a branched actin network and DdMyo7 is mostly autoinhibited in the cytoplasm. DdMyo7 monomers cycle back to a closed state if they do not dimerize. (**B**) VASP polymerizes actin at the leading edge, organizing the filaments into linear, parallel bundles. The DdMyo7 autoinhibitory state is destabilized is in the presence of this dense network via binding to actin filaments. (**C**) The motor of autoinhibited DdMyo7 binds to actin within the VASP-actin network, and the tail undergoes partner-mediated dimerization due to close proximity of other myosins. (**D**) Cooperative actions of VASP (bundles and polymerizes) and DdMyo7 dimers (bundle) organize actin filaments into nascent filopodia that continue to elongate by actin polymerization.

Interestingly, evidence for VASP-dependent changes in the cortical actin network has recently been shown in B16 melanoma cells (*Damiano-Guercio et al., 2020*). Their cortical actin network is normally dense and with many filaments oriented perpendicular to the membrane but in the absence of VASP, the network is more disperse and actin filaments are oriented in shallower angles with respect to the membrane (*Damiano-Guercio et al., 2020*). Thus, the formation of a VASP-mediated actin network at the cortex is likely to be important in this context as well as in other cell types.

## Motor-dependent release of autoinhibition and leading edge targeting

The motor domain of DdMyo7 and not the tail localizes this myosin to the dynamic leading edge of the cell (*Figure 1*). The tail alone is uniformly localized all around the cortex suggesting that it only provides general cortical targeting while the motor-FD is enriched at the front of the cell (*Figure 1*). Cortical targeting requires DdVASP or activated formin activity, suggesting an enrichment of parallel actin filaments in the cortex is needed for normal DdMyo7 activation (*Figure 3D*, *Figure 4D*). Mutation of DdMyo7 to lock the motor in a weak actin-binding state, even in the absence of autoinhibition such as in the case of the double non-hydrolyzer/autoinhibition mutant (E386A; KKAA), does not promote leading edge targeting. In contrast, a mutant that can bind actin but uncouples force generation from ATP hydrolysis (I426A) is localized more similarly to wild-type DdMyo7 (*Figure 7A–C*). Together these observations reveal that a functional motor that can hydrolyze ATP and bind actin is necessary for targeted, cortical recruitment of DdMyo7 to a DdVASP generated actin network, predominantly at the leading edge of the cell (*Figure 7*).

The motor is typically sequestered by interaction with the tail in the autoinhibited state, a widely used mechanism for controlling myosin activity and reducing unnecessary energy expenditure (*Heissler and Sellers, 2016*). DdMyo7 head/tail autoinhibition is mediated by charged residues in the FERM domain, similar to fly myosin 7a, and this regulates both cortical targeting and filopodia number (*Petersen et al., 2016*; *Arthur et al., 2019*; *Yang et al., 2009*). Preventing autoinhibition (KKAA) results in enhanced targeting to actin and, in the presence of DdVASP, filopodia formation (*Petersen et al., 2016*; *Arthur et al., 2019*; *Figure 7B,C,E,F*). It also bypasses the loss of cortical targeting observed in the absence of DdVASP, as the tail is now free to bind to any target sites in the cortex (*Figure 6*). The autoinhibited off state occurs when the equilibrium is biased toward the closed conformation. In this state, the motor has reduced affinity for actin and the partner binding sites in the tail are occluded. The results here lead to the following appealing model for how the motor specifies targeting of DdMyo7 to the leading edge (*Figure 8*). The compact, autoinhibited DdMyo7 freely diffuses throughout the cell. High local concentrations of actin in the cortical region facilitate interactions with actin, which destabilizes the inactive myosin, shifting the conformational equilibrium toward the open state. This makes the tail available to interact with partners and dimerize, and allows an active motor domain to engage with the actin network. If the motor does not dimerize, then DdMyo7 likely returns to a closed, off state, which diffuses and can be recycled back into the cytoplasm. It is tempting to speculate that the closely apposed actin filament bundles generated by VASP (or dDia2) favor stabilizing the open state because of the high local concentration of actin. This would promote dimerization, either because the tails could interact with each other more readily or because a dimerization partner is also present in the network. Once dimerized, this likely processive motor can productively engage with the cortex (e.g. generating force to align and/or bundle filaments) in collaboration with VASP to promote filopodia extension. Thus, motor domain mediated binding to a specific actin network is a key means of restricting localization of DdMyo7 during filopodia formation.

## Conserved and divergent models of filopodia myosin function

The mechanism of autoinhibition via motor-tail stabilizing interactions has been conserved throughout the evolution of myosins (*Umeki et al., 2011*; *Sakai et al., 2011*; *Yang et al., 2009*; *Petersen et al., 2016*; *Weck et al., 2017*; *Heissler and Sellers, 2016*). Mechanisms to overcome autoinhibition likely developed to restrict the recruitment and activation of myosins in time and space, allowing cells to fine-tune their activity. Here it is shown that DdMyo7 autoinhibition is relieved by the cortical actin network generated by DdVASP. While this differs from the PIP3-mediated recruitment of metazoan Myo10, once activated these two evolutionarily distant filopodia myosins use similar mechanisms to generate filopodia. They both dimerize upon recruitment to the

actin-rich cortex (*Arthur et al., 2019*; *Lu et al., 2012*) where they likely contribute to the reorganiza-tion of the Arp2/3 branched actin network to orient actin filaments perpendicular to the membrane (*Svitkina et al., 2003*; *Tokuo et al., 2007*; *Ropars et al., 2016*; *Arthur et al., 2019*). Interestingly, both DdMyo7 and Myo10 work in cooperation with VASP, although they appear to do so in different ways. In the case of Myo10, it is seen to co-transport with VASP along the length of filopodia during extension and co-immunoprecipitate with VASP, although it is not known if the two proteins interact directly in vivo (*Tokuo and Ikebe, 2004*; *Kerber et al., 2009*; *Lin et al., 2013*). In contrast, the activ-ity of DdVASP generates an actin-rich leading edge that recruits and activates DdMyo7 and there is no evidence that the two proteins interact with each other at present (*Figures 1*, *3* and *6*, and *Fig-ure 3—figure supplement 1A*).

*Dictyostelium* filopodia have actin filaments that are loosely bundled (*Medalia et al., 2007*). They lack an obvious fascin homologue, a bundling protein localized to filopodia in many cell types. How-ever, there is a calcium-sensitive bundler ABP-34 with three actin binding sites, similar to fascin (*Fechheimer, 1987*; *Lim et al., 1999*). ABP-34 is selectively localized to filopodia and has been sug-gested to play a role in filopodia formation in *Dictyostelium* but its exact function remains unclear (*Fechheimer, 1987*; *Rivero et al., 1996*). In the absence of a clear filopodia bundling protein, it is possible that DdMyo7 dimers contribute to actin bundling.

*Dictyostelium* are highly motile cells with dynamic, short lived filopodia. Given the early origins of Amoebozoa, a simple mechanism of coupling motor activation to actin dynamics at the pseudopod suggests that amoeboid filopodia formation is driven by a minimal regulatory circuit. It is tempting to speculate that VASP-dependent recruitment of an MF myosin represents an early form of cooper-ation between these two proteins. Perhaps as filopodia played wider roles in development and migration as multicellularity evolved, a signaling-based mechanism of MF myosin recruitment emerged with PIP3 binding to the three PH domain motif of Myo10 that replaced the first MF domain in present in other MF myosins. Thus, once Myo10 activation became dependent on PIP3, its motor activity was then used to promote VASP transport and filopodia extension. The evolution of the functional relationship between VASP and MF myosin would be interesting to explore in organisms such as *Drosophila* that make filopodia but lack Myo10 (*Drosophila* instead have a Myo22 with two MF domains like DdMyo7) and in the earliest organisms that have Myo7, Myo22, and Myo10 such as the filasterean *Capsaspora* and choanoflagellate *Salpingoeca* (*Kollmar and Mühlhau-sen, 2017*). It is possible that VASP plays a role in the recruitment of Myo10 to initiation sites or its activation in some systems, but this remains to be determined. The development of genetic tools in several evolutionarily significant organisms such as *Capsaspora* and *Salpingoeca* (*Parra-Acero et al., 2018*; *Booth et al., 2018*; *Booth and King, 2020*), unicellular organisms at the onset of multicellu-larity, should now allow for the study of the evolution of the VASP-filopodial MF myosin relationship in the targeting and activation of filopodial myosins.

## Materials and methods

### Cell lines, cell maintenance, and transformations

*Dictyostelium* control/wild-type (AX2 or AX3), *myo7* null (HTD17-1) (*Tuxworth et al., 2001*), *vasp* null (*Han et al., 2002*), and *dDia2* null (*Schirenbeck et al., 2005*) cells were cultured in HL5 media (Formedium). Transgenic lines were generated by methods similar to those described (*Gaudet et al., 2007*; *Paschke et al., 2018*). Briefly, axenic lines grown in HL5 cells were harvested, washed twice with ice cold H50 (20 mM HEPES, pH 7.0, 50 mM KCl, 10 mM NaCl, 1 mM MgSO$_4$, 5 mM NaHCO$_3$, 1 mM NaH$_2$PO$_4$ and flash spun at 8,000–10,000 X g in a microcentrifuge tube until the rotor reached speed). Cells were resuspended at 5e7 cells/mL and 100 µL of cells was combined with 10 or 15 µg DNA in a 0.1 cm gap cuvette. Cells were electroporated by pulsing twice, 5 s apart with a Bio-Rad Gene-Pulser set to 0.85 kV, 25 µF, and 200 Ω. Cells were recovered 10 min on ice and plated in a 10 cm dish for 24 hr before moving to selection media, either 10 µg/mL G418, 35 µg/mL HygromycinB or both.

Expression of the fusion proteins or null backgrounds was verified by western blotting using either anti-Myo7 (UMN87, *Tuxworth et al., 2005*), anti-GFP (Biolegend - B34), or anti-VASP (*Breitsprecher et al., 2008*) with anti-MyoB used as a loading control (*Novak et al., 1995*; *Figure 1—*

*figure supplement 2*). Bands were detected using fluorescent secondary antibodies (LiCor) imaged with the LiCor Odyssey.

## Generation of expression plasmids

The GFP-DdMyo7 expression plasmid was created by fusing *gfp* to the 5' end of the *myoi* gene (*Titus, 1999*) (dictyBase DDB: G0274455; Dictybase.org [*Fey et al., 2013*]) then cloning it into the pDXA backbone with the actin-15 promotor and a NeoR cassette as described (*Tuxworth et al., 2001*). Plasmids used for the expression of the full length tail (aa 809 - end) (*Tuxworth et al., 2001*), the KKAA autoinhibition mutant (K2333A/K2336A) (*Petersen et al., 2016*), motor forced dimer (aa 1–1020 followed by the mouse Myo5A coiled coil region and a GCN4 leucine zipper) *Arthur et al., 2019* have been described previously. An expression clone for the full-length GFP-DdMyo7 with a C-terminal prenylation site (CAAX) was generated using Q5 mutagenesis (New England Biolabs) to add codons encoding the CTLL* prenylation motif from *Dictyostelium* RasG (UNIPROT: P15064) to the 3' end of the *myoi* gene. A DdMyo7-mCherry expression plasmid was generated by first TA cloning a PCR product (myo42 to myi185 +2) encompassing the 3' region of the *myoi* gene (encoding aa 475 - end) minus the stop codon using StrataClone (Agilent) (pDTi289 +2). This fragment was then cloned into pDMCCherry, a modified pDM358 (*Veltman et al., 2009*) with the mCherry gene inserted for C-terminal fusions, generating pDTi299. The 5' end of *myoi* was then inserted by restriction cloning to create pDTi340. A motor mutant that cannot hydrolyze MgATP, the non-hydrolyzer E386V was designed based on a characterized *Dictyostelium* Myo2 mutant (*Friedman et al., 1998*). The combined non-hydrolyzer + KKAA was made by standard ligation cloning to introduce the motor domain sequence from the non-hydrolyzer mutant into KKAA full-length expression plasmid by restriction enzyme digest with BsiWI and BstEII. The uncoupler mutant (I426A) based on a characterized *Dictyostelium* Myo2 mutant (*Sasaki et al., 2003*) was cloned by Q5 mutagenesis. Fluorescent protein fusions of DdMyo7 were made using a combination of Q5 mutagenesis, Gibson assembly and restriction enzyme cloning. A DdMyo7-Scarlet I expression plasmid, pDTi517, was generated by first cloning a full-length *myoi* gene that has a BglII site at the 5' end, lacks its internal BglII site and also its 3' stop codon, pDTi515 +2, using a combination of Q5 mutagenesis, Gibson assembly and restriction enzyme cloning. The base Scarlet I-pDM304 expression plasmid for C-terminal fusions was generated by restriction enzyme cloning a codon-optimized synthesized Scarlet I gene (GenScript) into the extrachromosomal expression plasmid pDM304 (*Veltman et al., 2009*). The *myoi* gene was then cloned into mScarlet I-pDM304. Restriction cloning was used to create wild type and non-hydrolyzer mNeon DdMyo7 expression plasmids (pDTi516 and pDTi527, respectively). First, the base mNeon-pDM304 expression plasmid for N-terminal fusions was generated by restriction enzyme cloning a codon-optimized synthesized mNeon gene (GenScript) into the extrachromosomal expression plasmid pDM304 (*Veltman et al., 2009*). Then a full-length *myoi* gene that has a BglII site at the 5' end and lacks its internal Bgl II site was generated and cloned into either pDM448 (*Veltman et al., 2009*) for GFP-DdMyo7 expression (pDTi492) or mNeon-pDM304 for mNeon-DdMyo7 expression (pDTi516). Restriction cloning was used to exchange the 5' region of the gene carrying the E386V mutation into the wild type pDTi516 expression plasmid, creating pDTi527. The dDia2-CA mutant was created by first PCR-amplifying the *forH* gene (using forH4L and forH8 oligos) from GFP-dDia2 (*Schirenbeck et al., 2005*) then TA cloning the product using StrataClone (Agilent) to generate dDia2-SC. Mutations to generate a double R1035A, R1036A mutation were introduced dDia2-SC by Q5 mutagenesis (dDia2-CA SC). The wild type or mutant genes were restriction cloned into the extrachromosomal expression plasmid pDM449 (*Veltman et al., 2009*) to generate mRFPmars-dDia2-CA. An inducible V-1 expression plasmid was created by first PCR-amplifying the *mtpn* gene with V-1 F and V-1 R oligos (dictyBase:DDB_G0268038) using Ax2 genomic DNA then TA cloning the product using StrataClone (Agilent) to generateV-1 SC. The V-1 insert was then restriction cloned into the extrachromosomal expression plasmid pDM334 (*Veltman et al., 2009*) to generate GFP-inducible V-1. The sequence of all PCR generated clones was confirmed by Sanger sequencing (GeneWiz and University of Minnesota Genomics Center).

The GFP-VASP expression plasmid was a gift from Dr. Richard Firtel (UCSD) (*Han et al., 2002*). The VASP tetramer and FAB, Δtet mutants were not fused to GFP to avoid any steric hindrance with the fluorescent protein. The full-length VASP cDNA (dictyBase:DDB_G0289541) was cloned into the pDM344 shuttle vector (*Veltman et al., 2009*) and the NgoM-IV fragment from this plasmid was ligated into pDM358-mApple that has the mApple gene (*Shaner et al., 2008*) cloned in between

the *act6* promoter and *act15* terminator of pDM358 (*Veltman et al., 2009*). VASP-Δtet was created by introducing a SmaI site and stop codon into the *vasp* gene, altering the coding sequence from 334 PSLSAPL to 334 PSLSAPG* using Q5 mutagenesis. The F-actin-binding mutant (FAB K-E) was based on mutating previously identified critical F-actin binding residues (K275, R276, K278, and K280); (*Schirenbeck et al., 2006*) to glutamic acid (*Hansen and Mullins, 2010*) by Q5 mutagenesis. The sequence of all PCR generated clones was confirmed by Sanger sequencing (University of Minnesota Genomics Center). Oligonucleotides used are in the key resource table.

## Microscopy and imaging experiments

### Live-cell imaging

Microscopy of live cells was carried out as previously described (*Petersen et al., 2016*). Briefly, cells were adhered to glass bottom imaging dishes (CellVis, D35-10-1.5-N) and starved for 45–75 min in nutrient-free buffer (SB, 16.8 mM phosphate, pH 6.4), and then imaged at 1–4 Hz on a spinning disk confocal (3i Marianas or Zeiss AxioObserver Z.1) with a 63 X / 1.4NA Plan Apo oil-immersion objective. The sample temperature was maintained at 19–21°C. Samples were illuminated with 50 mW lasers (488 nm or 561 nm) with Semrock filters for 488- or 561-nm excitation and a Yokogawa CSU-X1 spinning disk, and captured with a Photometrics Evolve 512 EMCCD camera (final 0.212 micron pixel size). 4–6 Z sections of 0.28–0.5 microns were taken with a 50–250 ms exposure with 10–40% laser power. Cells were imaged for 10 s – 10 min or longer depending on experiment. Cells were plated at a density of $5 \times 10^5$ per mL. Ten fields of view were collected from each imaging dish, with 2–20 cells per field of view. All data sets represent cells from at least three independent experiments and two independently transformed cell lines.

### Drug treatments

Cells were washed free of media, adhered to glass bottom dishes and starved in nutrient-free buffer for 40 min. The buffer was replaced by buffer supplemented with the noted concentration of jasplakinolide (in 0.5% DMSO), cytochalasinA, latrunculin A, CK666, nocodazole, LY294002, wortmannin or just DMSO alone. Jasplakinolide treatment was for 5–20 min, cells were incubated with all other compounds for 15–20 min, prior to imaging for 10–30 min. Additional drug concentration data are in *Table 1*. Cells expressing mApple-DdMyo7 and inducible GFP-V1 were induced overnight with 10 µg/mL doxycycline (Sigma) to turn on expression of V-1 prior to imaging as above. Cells were treated with 0.25 µg/mL FM4-64 (Invitrogen) for 2–5 min to image filopodia with a membrane marker.

### LatrunculinA-induced actin waves

Cells expressing GFP-DdMyo7 and the actin reporter RFP-LimEΔcoil (*Gerisch et al., 2004*) were induced to generate travelling actin waves using a modified protocol (*Gerisch et al., 2004*). Cells rinsed with 16 mM phosphate buffer pH 6.4 were seeded on glass bottom dishes (Celvis) at $5 \times 10^5$ cell/mL, incubated in phosphate buffer for 30 min and then supplemented with 5 µM latrunculinA for 20 min. The solution was diluted to 0.5 µM latA and cells incubated for an additional 30 min, then imaged for up to 2 hr. Images were captured in 5–10 0.3 µm Z sections by spinning disk confocal microscopy (see above) every 5 s to make 10–30 min movies.

### Actin intensity linescans

Cells were seeded as above and fixed using picric acid (*Humbel and Biegelmann, 1992*). Cells were incubated with 568- or 647- Alexa phalloidin (Invitrogen) for 45 min, rinsed with PBS-glycine, then water and mounted using ProLong-Diamond (Molecular Probes). Slides were imaged using a Nikon Widefield Eclipse NiE microscope with a 40X/1.3 NA Plan Fluor oil objective, CoolSNAP ES2 CCD camera and SOLA light source. Maximum Z projections (5–10 0.3 µm) were analyzed by manually drawing a linescan perpendicular to the long axis and compiled with the FIJI/ImageJ macro and RStudio scripts (*Zonderland et al., 2019*).

### Cytoskeleton association

Log phase cells expressing wildtype or E386V mNeon-DdMyo7 were grown on bacteriological plastic plates (150 mm) were rinsed twice in PB then resuspended, counted and $5 \times 10^7$ cells collected

by centrifugation (Beckman J6, 200 x g). The pellet was resuspended in 1 ml lysis buffer (100 mM Tris, pH 8.1, 5 mM $MgCl_2$, 5 mM EDTA and 2.5 mM EGTA), washed once then lysed with Lysis Buffer +1% Tx-100, 1 mM TLCK (Sigma), 1 mM TPCK (Sigma), 1X HALT protease inhibitor cocktail (Pierce ThermoFisher) at room temperature. The 0.5 ml sample was spun immediately at 20,000 x g, 4 ℃ for 20 min. The supernatant was collected and pellet resuspended and homogenized in 0.25 ml of lysis buffer. An equal volume of ULSB gel sample buffer was added to aliquots from the sup and pellet and the samples run on a 4–15% gradient TGX SDS PAGE gel (BioRad) then transferred to nitrocellulose (Licor). The blot was probed for the presence of DdMyo7 and actin followed by fluo-rescent secondary antibodies (LiCor) and then imaged with the LiCor Odyssey. Quantification of the DdMyo7 band was performed using Image Studio Lite (Licor).

## Data analysis

### Protein alignments

Myosin motor domain sequences were aligned using T-Coffee algorithm 'Expresso' (*Notredame et al., 2000*). These included Uniprot sequence entries: P54697, Q9U1M8, P08799, P19524, K4JEU1, Q9V3Z6, Q9HD67,Q13402, Q6PIF6. Diaphanous related formins were aligned using Clustal Omega, using Uniprot entries: Q54N00, O70566, O60879, P41832, P48608 and the DAD domain basic residue highlighted is based on *Wallar et al., 2006*; *Lammers et al., 2005*. VASP mutations were made by first creating a structural alignment with DdVASP (Uniprot: Q5TJ65) and Human VASP (Uniprot: P50552) in T-Coffee algorithm 'Expresso'.

### Image analysis

Cortex to cell ratio, cortical asymmetry and filopodia per cell were quantified using a custom FIJI plugin 'Seven' (*Petersen et al., 2016*, code available on github linked to titus.umn.edu). Raw data for all experiments is provided (see source data files associated with each figure and table). Cells not expressing transgenic proteins were excluded from the analysis. Analysis was done on maximum intensity projections. First, the image is thresholded to mask the cell. This mask excludes the nucleus, which is devoid of DdMyo7 signal, and filopodia tips extended from the cell body. The cor-tical band (0.8 μm) intensity, standard deviation of intensity in the cortical band was measured for each cell. Next, a radial tip search identifies filopodia and registers them to each cell to count the number of filopodia per cell. Extending pseudopodia and actin correlation were done in FIJI by reslicing through extending pseudopodia and making plot profiles of the edge of the cell. Intensity profiles were normalized between 0 and 1, and multiple cells were averaged by fitting a restricted cubic spline.

## Statistical analysis

Statistics were calculated in Prism 8 (GraphPad). One-way ANOVA analysis with post hoc Tukey test or Dunnett's multiple comparison to wild-type control was used to compare groups; Student's t test was used when only comparing two datasets. Statistical tests were calculated on full datasets, exper-imental means are shown on graphs to demonstrate experimental variability. Data points deemed definite outliers (0.1%) by Rout method were excluded for cortex:cytoplasm ratio. Error bars are SEM, unless noted. Significance differences are in comparison to control (DdMyo7), unless noted. Tables have filopodia number as average number of filopodia in cells with at least one filopod. SEM is standard error of the mean. Capitalized 'N' indicates number of experiments lowercase n is num-ber of cells. Full statistical analysis is included in the source data tables accompanying each figure.

## Acknowledgements

We thank Livia Songster, Annika Schroder and Casey Eddington for many helpful discussions and assistance with experiments and analysis. We also thank Dr. Karl Petersen for the initial characteriza-tion of the motor mutations and for support with SEVEN. We are grateful to Professor Robert Insall (Beatson) for stimulating discussions, Dr. John Cooper (Washington U), Dr. Volodya Gelfand (North-western U), Dr. Lil Fritz-Laylin and members of the Fritz-Laylin lab (UMass Amherst) for helpful com-ments and critical reading of the manuscript. Thanks to Dr. Jan Faix (U Hannover) for providing the dDia2 nulls and VASP antibody and Dr Brad Nolen (U Oregon) for supplying the CK666, Dr. Günther

Gerisch (Max Planck) for the GFP-tubulin plasmid, Dr. GW Gant Luxton for use of the spinning disk confocal microscope and also to the University of Minnesota Imaging Center for additional imaging support. Our thanks to Dictybase.org for providing the Veltman plasmids and maintaining this valuable community resource. The AH team is part of the LabexCelTisPhyBio:11-LBX-0038, which is part of the Initiatives of Excellence of Université Paris Sciences et Lettres (ANR-10-IDEX-0001-02PSL). This work was supported by the CNRS, ANR-17-CE11-0029-01, and ANR-19-CE11-0015-02 (AH) and the NIH National Institute of General Medical Sciences (F31GM128325 to ALA and R01GM122917 to MAT).

# Additional information

## Funding

| Funder | Grant reference number | Author |
|---|---|---|
| Centre National de la Recherche Scientifique | | Anne Houdusse |
| Agence Nationale de la Recherche | ANR-17-CE11-0029-01 | Anne Houdusse |
| Agence Nationale de la Recherche | LabexCelTisPhyBio 11-LBX-0038 | Anne Houdusse |
| Agence Nationale de la Recherche | ANR-10-IDEX-0001-02PSL | Anne Houdusse |
| National Institutes of Health | F31GM128325 | Ashley L Arthur |
| National Institutes of Health | R01GM122917 | Margaret A Titus |

The funders had no role in study design, data collection and interpretation, or the decision to submit the work for publication.

## Author contributions

Ashley L Arthur, Conceptualization, Resources, Formal analysis, Validation, Investigation, Visualization, Methodology, Writing - original draft, Writing - review and editing; Amy Crawford, Resources, Formal analysis, Investigation; Anne Houdusse, Conceptualization, Formal analysis, Writing - review and editing; Margaret A Titus, Conceptualization, Resources, Formal analysis, Supervision, Funding acquisition, Investigation, Visualization, Writing - original draft, Project administration, Writing - review and editing

## Author ORCIDs

Ashley L Arthur https://orcid.org/0000-0001-8661-2873
Amy Crawford http://orcid.org/0000-0001-6147-7716
Anne Houdusse http://orcid.org/0000-0002-8566-0336
Margaret A Titus https://orcid.org/0000-0002-7583-9092

## Decision letter and Author response

Decision letter https://doi.org/10.7554/eLife.68082.sa1
Author response https://doi.org/10.7554/eLife.68082.sa2

# Additional files

## Supplementary files

• Transparent reporting form

## Data availability

All data generated or analyzed during this study are included in the manuscript and supporting files.

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

# Appendix 1

## Appendix 1—key resources table

| Reagent type (species) or resource | Designation | Source or reference | Identifiers | Additional information |
|---|---|---|---|---|
| cell line (*Dictyostelium*) | control wild type | Provided by Dr. Günther Gerisch (MPI Martinsried) | Ax2 | Available at Dictybase.org |
| cell line (*Dictyostelium*) | control wild type | Provided by Dr. Rick Firtel (UCSD) | Ax3 | Available at Dictybase.org |
| cell line (*Dictyostelium*) | *myo7* null | Titus Lab **Tuxworth et al., 2001** | *myo7* null HTD17-1 (G1-20) | Ax3 background |
| cell line (*Dictyostelium*) | control non-homologous recombinant - *myo7* null | Titus Lab **Tuxworth et al., 2001** | G1-21 | Ax3 background |
| cell line (*Dictyostelium*) | *vasp* null | Firtel Lab **Han et al., 2002** | *vasp*⁻ | Ax3 background |
| cell line (*Dictyostelium*) | dDia2 null | Provided by Dr. Jan Faix (Hannover Med Sch, Hannover Germany) **Schirenbeck et al., 2005** | *dia*⁻ | Ax2 background |
| Other | HL5 Medium including Glucose | Formedium | HLG0103 | *Dictyostelium* growth medium |
| Database | Dictybase.org | **Fey et al., 2013** | | |
| Software | Prism 8.0 | Graphpad | Statistical Analysis | graph preparation, statistical analysis |
| Software | Seven | **Petersen et al., 2016** | | https://github.com/tituslabumn/Seven |
| Antibody | Rabbit anti-DdMyo7 | UMN-87 **Tuxworth et al., 2005** | | 1:2000 |
| Antibody | mouse monoclonal anti-GFP, clone B34 | Biolegend | 902602 | 1:5000 |
| Antibody | Rabbit polyclonal anti-vasp | Provided by Dr. Jan Faix (Hannover Med Sch, Hannover Germany) **Breitsprecher et al., 2008** | | 1:500 |
| Antibody | Rabbit polyclonal anti myoB | Titus Lab **Novak et al., 1995** | | 1:2000 |
| Antibody | Rabbit polyclonal anti-mCherry | Proteintech | 26765–1-AP | 1:500 |
| Antibody | Goat anti rabbit secondary | Licor | IR680 | 1:2500-1:10,000 |
| Antibody | Goat anti mouse secondary | Licor | IR800 | 1:2500-1:10,000 |
| Antibody | Mouse monoclonal anti-*Dictyostelium* actin | Provided by Dr. Günther Gerisch (MPI Martinsried) **Westphal et al., 1997** | | 1:4000 |
| Chemical compound | G418 | Fisher Scientific | BP-673 | antibiotic |
| Chemical compound | G418 | Gold Biotechnology | G-418 | antibiotic |

*Continued on next page*

*Appendix 1—key resources table continued*

| Reagent type (species) or resource | Designation | Source or reference | Identifiers | Additional information |
|---|---|---|---|---|
| Chemical compound | Hygromycin B | Gold Biotechnology | H-270 | antibiotic |
| Chemical compound | Doxycycline Hyclate | Sigma | D9891 | antibiotic |
| Chemical compound | Pencillin G Sodium Salt | Sigma | P3032 | antibiotic |
| Chemical compound | Streptomycin Sulfate | Sigma | S9137 | antibiotic |
| Chemical compound, drug | Cytochalasin A | Sigma | C-6637 | |
| Chemical compound, drug | Jasplakinolide | Sigma | J4580 | |
| Chemical compound, drug | Latrunculin A | Sigma | L5163 | |
| Chemical compound, drug | LY294002 | EMD Millipore | 440204 | |
| Chemical compound, drug | Nocodazole | Sigma | M1404 | |
| Chemical compound, drug | Wortmannin | Sigma | W1628 | |
| Chemical compound, drug | CK666 | Gift from Dr. Brad Nolen (U. Oregon) *Nolen et al., 2009* | | |
| Chemical compound, drug | Alexa Fluor 568 Phalloidin | Invitrogen | A12380 | |
| Chemical compound, drug | Alexa Fluor 647 Phalloidin | Invitrogen | A22287 | |
| Chemical compound, drug | FM 4-64 Dye | ThermoFisher | T13320 | |
| other | Q5 polymerase | New England Biolabs | M049L | |
| commercial assay or kit | NEBuilder HiFi DNA Assembly | New England Biolabs | E5520S | |
| Recombinant DNA reagent | synthesized gene - mScarlet I | Genscript | actagtggtggttcaggaGTTTCAAAAGGTGAAGCCGTTATTAAAGAATTTATGAGATT CAAGGTTCACATGGAAGGAAGTATGAACGGTCATGAATTTGAGATTGAAGGAG AAGGTGAAGGTAGACCATATGAAGGCACCCAAACAGCTAAATTAAAAGTAACT AAAGGTGGTCCATTACCATTTAGTTGGGATATTTTATCTCCACAATTTATGTATGG TTCACGTGCTTTCAttAAACATCCAGCAGATATTCCAGATTATTATAAACAATCATT TCCAGAA GGTTTTAAATGGGAACGTGTCATGAACTTTGAAGATGGTGGAGCAGT TACAGTCACACAAGATACCTCATTAGAAGATGGTACATTAATATATAAAGTTAAAT TACGTGGTACTAATTTTCCACCAGACGGTCCAGTAATGCAAAAAAAACAATGGG CTGGGAAGCTAGT ACAGAACGTTTATATCCTGAAGATGGTGTCCTTAAAGGCGA TATAAAAATGGCCTTGAGATTAAAGGATGGTGGTAGGTATTTAGCAGATTTCAAA ACCACTTATAAAGCAAAAAAACCAGTTCAAATGCCAGGTGCATATAATGTTGATA GAAAACTTGATATTACCAGTCATAATGAAGATTACACAGTTGTCGAACAATACGAA CGTTCTGAAGGTCGTCATAGCACTGGTGGTATGGATGAATTATACAAATAAgctagc | |

*Continued on next page*

*Appendix 1—key resources table continued*

| Reagent type (species) or resource | Designation | Source or reference | Identifiers | Additional information |
|---|---|---|---|---|
| Recombinant DNA reagent | synthesized gene - mNeon | Genscript | ggatccATGGTGAGTAAAGGTGAAGAAGATAATATGGCATCGTTACCAGCTACACATGAG TTACATATATTCGGTAGCATTAATGGTGTTGATTTTGATATGGTGGGACAAGGTACCGGT AATCCTAATGATGGTTACGAAGAACTAAATTTAAAATCGACTAAAGGTGACTTACAATTT TCTCCATGGATTTTAGTGCCACATATAGGGTATGGTTTTCATCAATACTTACCATATCCAG ATGGTATGTCACCATTTCAAGCTGCAATGGTTGATGGATCAGGTTATCAAGTTCATAGAA CAATGCAATTTGAAGATGGTGCTTCATTAACTGTTAATTATAGATACACATATGAAGGCTC ACATATTAAAGGTGAAGCTCAAGTTAAAGGTACTGGTTTCCCAGCCGATGGCCCAGTTAT GACAAATAGTTTAACAGCAGCAGATTGGTGTAGATCCAAAAAAACTTATCCAAATGATAA AACAATTATTTCAACTTTTAAATGGTCATATACAACCGGTAATGGTAAACGTTATCGTTCAA CAGCCCGTACAACATATACTTTTGCTAAACCAATGGCAGCTAATTATTTAAAAAATCAACC AATGTATGTTTTTCGTAAAACAGAGTTAAAACATTCAAAAACAGAACTTAATTTTAAAGAAT GGCAAAAAGCATTTACAGACGTTATGGGTATGGATGAACTTTATAAGagatct | |
| Sequence-based reagent | CAAX F | IDTDNA | PCR Primer (pDTi346 plasmid) | ttattaTAAAAAAATTAAAA TAAAATAAAATCTCGTG |
| Sequence-based reagent | CAAX R | IDTDNA | PCR Primer (pDTi346 plasmid) | tgtacaTTGAGAAGAATAA AATTGATAAACTG |
| Sequence-based reagent | E386V F | IDTDNA | PCR Primer (pDTi364 plasmid) | ttttgtAAATTTTAAAAAGAAT AGTTTTGAACAATTTTG |
| Sequence-based reagent | E386V R | IDTDNA | PCR Primer (pDTi364 plasmid) | ccaaagATATCCAATAC ACCAATAAATGTTG |
| Sequence-based reagent | I426A F | IDTDNA | PCR Primer (pDTi435 plasmid) | AAAAGAAAAAgctAATTGG AGTAAGATCGTATATAATG |
| Sequence-based reagent | I426A R | IDTDNA | PCR Primer (pDTi435 plasmid) | TCATATTCTTCTTGTTCT AATTTAAAAATATG |
| Sequence-based reagent | myi42 | in-house synthesis | PCR Primer (pDTi289 + 2 plasmid) | catgccatggcagcagcagca ACCTTAAAGAGAAAAGCACCAGTCG |
| Sequence-based reagent | myi185 + 2 | IDTDNA | PCR primer (pDTi289 + 2 plasmid) | gctagcaaTTGAGAAGAA TAAAATTGATAAACTGAAGC |
| Sequence-based reagent | VASP339* F | IDTDNA | related to pVASP29 plasmid | taataaAGAGCATCTCAACATTAACTAG |
| Sequence-based reagent | VASP339* R | IDTDNA | PCR Primer (pVASP29 plasmid) | cccgggAGCTGATAAGGATGGTGAAG |
| Sequence-based reagent | FAB K-E F | IDTDNA | PCR Primer (pVASP34 plasmid) | gaaatggagGCAGCAGCATC TCAACCAA |
| Sequence-based reagent | FAB K-E R | IDTDNA | PCR primer (pVASP34 plasmid) | ggcttcctcGGCCATAACTTCGGCCAT- |
| Sequence-based reagent | forH4L F | IDTDNA | PCR primer (dDia2-SC plasmid) | AATTGACCAGATCTAATTTGAG |
| Sequence-based reagent | forH8 R | IDTDNA | PCR primer (dDia2 SC plasmid) | actagtTTATTTTTTTA ATTGGCCTGATGG |
| Sequence-based reagent | forH9 F | IDTDNA | PCR primer (dDia2-CA SC plasmid) | ggatccATGTCTTTTGATTTA GAGAGTAATAGTAGTGG |
| Sequence-based reagent | forH10 R | IDTDNA | PCR primer (dDia2-CA SC plasmid) | ATTCAAAGATagaagaGTT GGTGATTCTGTCATTG |
| Sequence-based reagent | V1 F | IDTDNA | PCR primer (V-1 SC plasmid) | agatctATGGAAGAA CAAAATGATTTCAC |
| Sequence-based reagent | V1 R | IDTDNA | PCR primer (V-1 SC plasmid) | actagtTTATTTTAATAA TGCTTTAATATCAGC |
| Transformed construct (*Dictyostelium*) | GFP-DdMyo7-Tail | Titus Lab *Tuxworth et al., 2005* | pDTi35 | pSmall, G418, Extrachromosomal |

*Continued on next page*

*Appendix 1—key resources table continued*

| Reagent type (species) or resource | Designation | Source or reference | Identifiers | Additional information |
|---|---|---|---|---|
| Transformed construct (*Dictyostelium*) | GFP-DdMyo7 | Titus Lab *Tuxworth et al., 2001* | pDTi74 | pBS, G418, Integrating |
| Transformed construct (*Dictyostelium*) | GFP-DdMyo7motor-FD | Titus Lab *Arthur et al., 2019* | pDTi490 | pDXA, G418, Extrachromosomal |
| Transformed construct (*Dictyostelium*) | GFP-DdMyo7-KKAA | Titus Lab *Arthur et al., 2019* | pDTi321 | pBS, G418, Integrating |
| Transformed construct (*Dictyostelium*) | GFP-DdMyo7-E386V | Titus Lab this paper | pDTi364 | pBS, G418, Integrating |
| Transformed construct (*Dictyostelium*) | GFP-DdMyo7-E386V; KKAA | Titus Lab this paper | pDTi386 | pBS, G418, Integrating |
| Transformed construct (*Dictyostelium*) | GFP-DdMyo7-CAAX | Titus Lab this paper | pDTi346 | pBS, G418, Integrating |
| Transformed construct (*Dictyostelium*) | GFP-DdMyo7-I426A | Titus Lab this paper | pDTi435 | pTX-GFP, G418, Extrachromosomal |
| Transformed construct (*Dictyostelium*) | GFP-DdMyo7 | Titus Lab this paper | pDTi504 | pDM317, G418, Extrachromosomal |
| Transformed construct (*Dictyostelium*) | mNeon-DdMyo7 | Titus Lab, this paper | pDTi516 | pDM304, G418, Extrachromosomal |
| Transformed construct (*Dictyostelium*) | mNeon-DdMyo7-E386V | Titus Lab, this paper | pDTi527 | pDM304, G418, Extrachromosomal |
| Transformed construct (*Dictyostelium*) | DdMyo7-mCherry | Titus Lab, this paper | pDTi340 | pDM358, Hyg, Extrachromosomal |
| Transformed construct (*Dictyostelium*) | mApple-DdMyo7 | Titus Lab | pDTi512 | pDM358, Hyg, Extrachromosomal |
| Transformed construct (*Dictyostelium*) | GFP-VASP | Provided by Dr. Rick Firtel (UCSD) *Han et al., 2002* | GFP-VASP | EXP4+, G418, Extrachromosomal |
| Transformed construct (*Dictyostelium*) | VASP | Titus Lab, this paper | VASP | pDM358, Hyg, Extrachromosomal |
| Transformed construct (*Dictyostelium*) | VASP-ΔTET | Titus Lab, this paper | VASP-ΔTET | pDM358, Hyg, Extrachromosomal |
| Transformed construct (*Dictyostelium*) | VASP-FAB | Titus Lab, this paper | VASP-FAB | pDM358, Hyg, Extrachromosomal |
| Transformed construct (*Dictyostelium*) | diaWT | Provided by Dr. Jan Faix (Hannover Med Sch, Hannover Germany) *Schirenbeck et al., 2005* | GFP-dDia2 | pDGFP-MCS, G418, Integrating |
| Transformed construct (*Dictyostelium*) | diaCA | Titus Lab, this paper | mRFP-mars-diaCA | pDM449, Hyg, Extrachromosomal |

*Continued on next page*

*Appendix 1—key resources table continued*

| Reagent type (species) or resource | Designation | Source or reference | Identifiers | Additional information |
|---|---|---|---|---|
| Transformed construct (*Dictyostelium*) | GFP-V1 | Titus Lab, this paper | Tetracycline - Inducible GFP-V1 | pDM334, G418, Extrachromosomal |
| Transformed construct (*Dictyostelium*) | RFP-Lifeact | *Brzeska et al., 2014* | | pDM358, Hyg, Extrachromosomal |
| Transformed construct (*Dictyostelium*) | RFP-LimEΔcoil | *Gerisch et al., 2004* | | |
| Transformed construct (*Dictyostelium*) | GFP-tubulin | Provided by Dr. Günther Gerisch (MPI Martinsried) *Neujahr et al., 1998* | | pDEXRH, G418, Integrating |

