## [Decision Letter]

**Acceptance summary:**

The data presented here support the hypothesis that actin architecture plays a role in the recruitment of actin polymerase VASP and the filopodia machinery and further illuminates the relationship between the myosin DdMyo7, DdVASP, and filopodial formation in *Dictyostelium*. Given the importance of filopodia formation to a myriad of cellular processes, this paper should draw broad attention from the cell biology community.

**Decision letter after peer review:**

Thank you for submitting your article "VASP mediated actin dynamics activate and recruit a filopodia myosin" for consideration by *eLife*. Your article has been reviewed by 3 peer reviewers, including Derek Applewhite as the Reviewing Editor and Reviewer #1, and the evaluation has been overseen by Anna Akhmanova as the Senior Editor. The following individual involved in review of your submission has agreed to reveal their identity: Scott Hansen (Reviewer #2).

Summary:

This paper will be of general interest to cell biology community as it focuses on the basic mechanisms of filopodia formation. The data presented here supports the claims by authors in showing that a specific actin architecture is needed to fully recruit filopodia machinery. Several quantitative imaging techniques were used to rigorously determine the requirements of this actin architecture.

Essential revisions:

1) Provide some discussion about the actin bundling protein, fascin, at the beginning and/or end of the manuscript. From my understanding, fascin is not expressed in Dicty. However, it's important for generating filopodia in many mammalian cell types. Do DdVASP and DdMyo7 serve the function of fascin? Alternatively, are there other actin bundling proteins that are critical for filopodia formation in Dicty?

2) There are a couple of far-reaching statements that should either be supported with more experiments or toned down with further explanation.

a. Line 195: "The microtubule disrupting compound nocodazole had no effect on DdMyo7 in spite of the presence of two MF domains with micromolar affinity for microtubules in the DdMyo7 tail." In Supp. Figure 2A, I wouldn't say that there's "no effect" because there is clearly less DdMyo7 in the nocodazole treated cell than in the control image.

b. In line with the point above, look at Line 190: "The changes in DdMyo7 localization are specific to treatments that alter actin dynamics." Considering there are these microtubule binding domains in the DdMyo7 tail, only a few microtubule based assays were performed. What about the effect of microtubule dynamics? What about different drug concentrations? I think probing the level of actin-microtubule crosstalk that could potentially be at play here would be fascinating, but also outside the scope of this paper. However, to back up the statement in Line 190 that is due to actin dynamics alone, either more experiments need to be performed, or the statement should be toned down while considering other mechanistic possibilities.

c. Line 238: "Actin waves were generated in either control or vasp null cells and in all cells visualized all waves are completely devoid of DdMyo7 (Figure 4A)." First of all, do you mean that the waves are devoid of DdMyo7 in both the control and vasp- cells? In the control of Figure 4A, there is very slight enhancement of DdMyo7 at the identified actin wave site. In the vasp- case, there is still DdMyo7 present in the location of the actin wave, but it just isn't concentrated in a ring type formation – so to say that it is completely devoid of it is not true. Please clarify.

3. Discussion section: Actin-dependent release of autoinhibition and leading edge targeting

In line 418, you say "Overcoming autoinhibition is thus essential for liberating the motor and allowing it to target DdMyo7 to the dynamic cortex."

Then in line 424, "Motor activity (i.e. ATP hydrolysis) and actin binding are needed to release autoinhibition and promote cortical recruitment of DdMyo7."

These statements seem circular to me. I agree with what is said in Line 418. But in Line 424, how can motor activity and actin binding release autoinhibition if autoinhibition prevents motor activity and actin binding?

You show that blocking autoinhibition results in enhanced cortical targeting and filopodia formation, and you say at the end of the paragraph that the motor domain mediated binding to a specific actin network is a key way to restrict localization of DdMyo7 during filopodia formation, but I don't think that necessarily points to the conclusion drawn in the section heading that autoinhibition is released by actin-binding.

---

## [Author Response]

Essential revisions:1) Provide some discussion about the actin bundling protein, fascin, at the beginning and/or end of the manuscript. From my understanding, fascin is not expressed in Dicty. However, it's important for generating filopodia in many mammalian cell types. Do DdVASP and DdMyo7 serve the function of fascin? Alternatively, are there other actin bundling proteins that are critical for filopodia formation in Dicty?

We appreciate your bringing up this important point. It is true that there does not appear to be a true fascin homolog in Dicty. ABP-34 is the best candidate for a filopodia actin bundling protein, it is a small, Ca^2+^-sensitive bundling protein with three actin binding sites that has been localized along the length of filopodia (Fechheimer, 1987, JCB). While it clearly bundles actin filaments, its role in filopodia formation has not been thoroughly investigated. Early work suggests that it regulates filopodia formation but the reported phenotypes of different knock-out mutants (Rivero et al., 1996, JCB) are hard to reconcile. The other known major bundling proteins, a-actinin and the filamin homologue (gelation factor) are not reported to be localized to filopodia and have not been implicated in their formation.

We now discuss the potential role of a known *Dictyostelium* actin bundler in filopodia formation and also propose a possible role of DdMyo7 dimers to serve a role in filament bundling (lines 486-492):

“*Dictyostelium* filopodia have actin filaments that are loosely bundled (Medalia et al., 2007). […] In the absence of a clear filopodia bundling protein, it is possible that DdMyo7 dimers contribute to actin bundling.”

2) There are a couple of far-reaching statements that should either be supported with more experiments or toned down with further explanation.a. Line 195: "The microtubule disrupting compound nocodazole had no effect on DdMyo7 in spite of the presence of two MF domains with micromolar affinity for microtubules in the DdMyo7 tail." In Supp. Figure 2A, I wouldn't say that there's "no effect" because there is clearly less DdMyo7 in the nocodazole treated cell than in the control image.

Thanks to the reviewers for their keen eye. The original experiments used DdMyo7-mCherry (to image with GFP-tubulin) which does not have as good of signal to noise as GFP fusion proteins, posing challenges for some of the analysis. Prompted by this reviewer's comment we revisited the nocodazole experiments and include a full new dataset testing the role of nocodazole on DdMyo7 cortical recruitment and filopodia formation at a range of drug concentrations using cells that only express mNeon-DdMyo7 that is much brighter than the original mCherry fusion. The impact of nocodazole on cells only expressing GFP-tubulin was analyzed in parallel to assess the effectiveness of the drug. In these experiments, treatment of cells with increasing concentrations of nocodazole (5, 15, 50 µM) progressively disassembles microtubules without affecting the overall DdMyo7 expression levels (Figure 2—figure supplement 1D). (more below)

b. In line with the point above, look at Line 190: "The changes in DdMyo7 localization are specific to treatments that alter actin dynamics." Considering there are these microtubule binding domains in the DdMyo7 tail, only a few microtubule based assays were performed. What about the effect of microtubule dynamics? What about different drug concentrations? I think probing the level of actin-microtubule crosstalk that could potentially be at play here would be fascinating, but also outside the scope of this paper. However, to back up the statement in Line 190 that is due to actin dynamics alone, either more experiments need to be performed, or the statement should be toned down while considering other mechanistic possibilities.

The reviewers raise an important point about microtubule and actin network cross-talk. It is true we have not carried out an extensive analysis of the role of microtubules or microtubule-actin crosstalk during filopodia formation or the role of microtubules in DdMyo7 function or localization. However, prompted by this comment and the one above we went back to take a closer look at the impact of microtubules on filopodia formation and DdMyo7 cortical localization.

We do not observe robust colocalization of DdMyo7 on microtubules, that typically emanated radially from the cell’s centrosome, as quantified by Cytofluorogram analysis (Figure 2—figure supplement 1D).

A full new dataset testing the role of nocodazole on DdMyo7 cortical recruitment and filopodia formation at a range of drug concentrations using cells that only express mNeon-DdMyo7 is now included in the manuscript. The effect of nocodazole on cells only expressing GFP-tubulin was analyzed in parallel to assess the effectiveness of the drug. Treatment of cells with increasing concentrations of nocodazole (5, 15, 50 µM) is seen to progressively disassemble microtubules without affecting the overall DdMyo7 expression levels (Figure 2—figure supplement 1E). The cells do appear smaller and rounder at higher concentrations where microtubule depolymerization is most complete. Nocodazole does not appear to eliminate filopodia formation altogether as cells continue to make filopodia when the microtubule network is depolymerized. However, the average filopodia number decreases with increasing nocodazole concentration (Figure 2—figure supplement 1I; Table 1). We do see a significant disruption in cortical recruitment of mNeon-DdMyo7 treated with 5-50µM Nocodazole compared to the DMSO control. Interestingly, 30 µM nocodazole treatment does not reduce the cortical localization of the GFP-DdMyo7 tail fragment (Figure 2—figure supplement 1F,H), suggesting that the reduced cortical localization of DdMyo7 is due to a change in actin dynamics and not because its localization is directly microtubule dependent.

In light of the new nocodazole results, we cannot so simply state that actin dynamics alone are all that is needed to control DdMyo7 localization and that microtubules have no role. Indeed, it would be interesting in the future to investigate how microtubule-actin crosstalk might impact actin modulating proteins or actin dynamics that are required for DdMyo7 recruitment. Given this, we have now include the new data in Figure 2—figure supplement 1, and add the text quoted below (lines 193-204) and, importantly, have toned down the language.

“Treatment of *Dictyostelium* cells with the microtubule depolymerizing compound nocodazole disrupts cytosolic microtubules (visualized by cells expressing GFP-tubulin (Neujahr et al., 1998), Figure 2 – figure 2 supplement 1E). […] Together, the data suggest that microtubule-actin crosstalk may impact the dynamics of the cortical actin network and thus indirectly control the cortical recruitment of full length DdMyo7.”

c. Line 238: "Actin waves were generated in either control or vasp null cells and in all cells visualized all waves are completely devoid of DdMyo7 (Figure 4A)." First of all, do you mean that the waves are devoid of DdMyo7 in both the control and vasp- cells? In the control of Figure 4A, there is very slight enhancement of DdMyo7 at the identified actin wave site. In the vasp- case, there is still DdMyo7 present in the location of the actin wave, but it just isn't concentrated in a ring type formation – so to say that it is completely devoid of it is not true. Please clarify.

We apologize for the overstatement here. What we mean is that unlike PIP3 or Myo1B or Myo1C (such as shown in Brzeska et al. Plos One 2014), there is no strong co-localization of DdMyo7 to the traveling actin wave. To more clearly demonstrate this, we have included an example linescan through a kymograph that shows the peaks of actin intensity during a wave that does not coincide with peaks of DdMyo7 localization (Figure 4—figure supplement 1B). We have also updated the text so that it now states (lines 262-266):

“Control or vasp null cells expressing both RFP-LimE (a marker for F-actin) and GFP-DdMyo7 were induced to form waves that are readily apparent as broad circles of actin that emanate outwards towards the cell periphery. Robust actin waves marked by RFP-LimE were observed but DdMyo7 was not observed in the waves in either cell line (n=6 per genotype; Figure 4A, Figure 4—figure supplement 1A,B).”

3. Discussion section: Actin-dependent release of autoinhibition and leading edge targetingIn line 418, you say "Overcoming autoinhibition is thus essential for liberating the motor and allowing it to target DdMyo7 to the dynamic cortex."Then in line 424, "Motor activity (i.e. ATP hydrolysis) and actin binding are needed to release autoinhibition and promote cortical recruitment of DdMyo7."These statements seem circular to me. I agree with what is said in Line 418. But in Line 424, how can motor activity and actin binding release autoinhibition if autoinhibition prevents motor activity and actin binding?

We appreciate your pointing out the inconsistency between the two statements, highlighting the need to make the key points in the discussion clear. This section of the Discussion has now been revised and expanded a bit to better describe our current view of how autoinhibition is released. We would like to emphasize that the autoinhibited conformation is in an equilibrium with the open, active conformation but largely biased towards the closed state in the cytosol. In this state, the motor has weak affinity for actin. Once in regions of high local concentration of actin, interaction of the motor with actin would destabilize the closed, inhibited conformation to switch it towards the open state. The motor could then bind to actin and dimerize. Put another way, actin binding is not directly relieving autoinhibition but rather it is necessary for myosins to be captured in their open state when the dense local actin network is comprised mainly of parallel actin filaments. The exact mechanism of this mechanism remains unknown and a major goal for future experiments will be to determine the precise order of events and relationship between autoinhibition and motor activity.

The following text has been added to the Discussion (line 448-463) to better clarify our model and also indicate the uncertainty as the precise nature of these steps -

“The results here lead to the following appealing model for how the motor specifies targeting of DdMyo7 to the leading edge (Figure 8). […] Thus, motor domain mediated binding to a specific actin network is a key means of restricting localization of DdMyo7 during filopodia formation.”

You show that blocking autoinhibition results in enhanced cortical targeting and filopodia formation, and you say at the end of the paragraph that the motor domain mediated binding to a specific actin network is a key way to restrict localization of DdMyo7 during filopodia formation, but I don't think that necessarily points to the conclusion drawn in the section heading that autoinhibition is released by actin-binding.

We appreciate this comment and to better represent our findings we have changed the section heading from: “Actin dependent release of autoinhibition and leading edge targeting” to “Motor dependent release of autoinhibition and leading edge targeting”.